# DynaGuide: Steering Diffusion Polices with Active Dynamic Guidance

**Maximilian Du**
Stanford University
maxjdu@stanford.edu

**Shuran Song**
Stanford University
shuran@stanford.edu

**dynaguide.github.io**

## Abstract

Deploying large, complex policies in the real world requires the ability to steer them to fit the needs of a situation. Most common steering approaches, like goal-conditioning, require training the robot policy with a distribution of test-time objectives in mind. To overcome this limitation, we present DynaGuide, a steering method for diffusion policies using guidance from an external dynamics model during the diffusion denoising process. DynaGuide separates the dynamics model from the base policy, which gives it multiple advantages, including the ability to steer towards multiple objectives, enhance underrepresented base policy behaviors, and maintain robustness on low-quality objectives. The separate guidance signal also allows DynaGuide to work with off-the-shelf pretrained diffusion policies. We demonstrate the performance and features of DynaGuide against other steering approaches in a series of simulated and real experiments, showing an average steering success of 70% on a set of articulated CALVIN tasks and outperforming goal-conditioning by 5.4x when steered with low-quality objectives. We also successfully steer an off-the-shelf real robot policy to express preference for particular objects and even create novel behavior. Videos and other visualizations can be found on the project website: https://dynaguide.github.io

## 1 Introduction

The rise of large datasets and expressive policies has enabled complicated skills in robots like folding clothes [3, 55], making sandwiches [44], and washing dishes [7]. When these large policies are deployed in the dynamic real world, we face the challenge of *steering* them to match the needs of a specific scenario. This means finding a subset of the policy's behaviors that are appropriate for the scenario [48]. The policy's complexity means that we ideally want to accomplish this steering without needing to retrain the policy, sample excessively from it, or anticipate all the possible steering during policy training.

Most existing works for policy steering, however, rely on at least one of those assumptions. Even language or goal-conditioned policies are trained on a set of

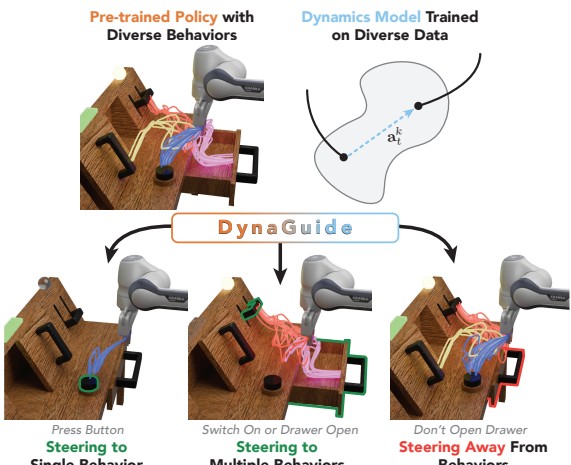

Figure 1: **DynaGuide** steers **pretrained diffusion policies** by adding guidance from a **dynamics model** into the action denoising process. This dynamics-based guidance can take a diverse behavior base policy and steer it towards one single behavior (left), multiple behaviors (middle), and even removing a behavior (right)— all without fine-tuning.

39th Conference on Neural Information Processing Systems (NeurIPS 2025).

goal distributions, which requires foresight for the goal distribution that satisfies all future steering needs [45]. In our work, we distance ourselves from these assumptions. We propose *separating* the steering forces from the base behavior policy and *combining* the influence from the steering model with the strong action prior of the base policy to generate guided behaviors that satisfy inference-time objectives while leveraging the base policy's existing skills.

Steering robot behavior requires understanding how current actions influence future outcomes, a feat well-suited for dynamics models [39, 16]. To adapt a dynamics model for policy steering, we craft a latent visual dynamics model that predicts the final or far-future observation of a trajectory given the current observation and action, allowing it to understand long-horizon dependencies. This dynamics model learns from unstructured environment interaction data and operates on visual observations projected into the expressive DinoV2 latent space [37] previously shown to be useful for robot planning [56]. In this latent space, we can compute the distance between the predicted future observation and observations of other desired/undesired outcomes, creating a differentiable metric that encompasses the guidance objectives.

Previous methods that steer policies with external metrics will sample actions from the policy and pick the action that best satisfies the metric [39, 34]. Instead, our approach leverages the denoising stochastic process of diffusion-based policies. Taking inspiration from classifier guidance [9] and dynamics diffusion guidance in other applications [53], we take the gradient of the metric through the dynamics model and use this dynamics gradient to directly modify the action denoising process.

This separate dynamics model and the active diffusion guidance make up DYNAGUIDE, a new way of steering pretrained diffusion policies towards complex objectives. This design offers several key advantages over existing steering approaches:

- **Flexible Steering Structure.** Goal-conditioned formulations have rigid input structures that cannot be modified during inference–and most policies accept only a single goal condition. DynaGuide accepts a collection of guidance conditions encompassing any number of positive and/or negative objectives (§4.3).

- **Increased Steering Robustness.** Goal-conditioned approaches require training the input condition alongside the policy, causing out-of-distribution (OOD) failures when new or lower-quality objectives are introduced during inference. The DinoV2 embedder and separation of guidance and policy in DynaGuide means that no individual part of the system is OOD under this same situation, and the averaging effect of the guidance metric (Eq. 2) means that DynaGuide still extracts a meaningful guidance signal (§4.2). The additional experience present in the dynamics model also enables guidance towards novel behavior (§4.5).

- **Plug-and-Play Modularity.** Unlike fine-tuning approaches, DynaGuide only modifies the inference process of the diffusion policy, allowing different dynamics models to be swapped in and out without changing the base policy. This modularity also allows an *off-the-shelf* policy to be steered without further modifications (§4.5).

- **Enhancing Underrepresented Behaviors.** Sample-based steering approaches [39, 34] select the best action from a set of proposed actions by the base policy. Not only does this require many inferences of the policy per step, these sampling approaches also struggles with expressing behaviors that are underrepresented in the base policy (§4.4). In contrast, our Dynamics Guidance directly influences the denoising process. Through one denoising sequence, we guide the action towards the specified objectives, even if the behavior is less common in the base policy (§4.4).

To investigate these claims, we conduct five experiments using simulated CALVIN environment tasks [30] and three experiments on a real robot arm. We show that DynaGuide successfully guides the base policy up 70% of the time (§4.1) and outperforms goal conditioning by 5.4x on lower quality objectives (§4.2). It successfully amplifies underrepresented behaviors over sampling methods (§4.4) and accommodates multiple positive and negative objectives (§4.3). DynaGuide is also successful on an existing real robot policy, achieving up to 80% guidance success and even creating a novel behavior (§4.5). We will make code and collected data publicly available. Videos and visualizations can be found on our website, https://dynaguide.github.io

## 2    Related Works

**Conditioning Policies on Goal Representations.** A common approach that influences robot behavior is *goal conditioning*, where the policy is given a representation of a desired outcome. The common representation is natural language [5, 57, 23, 36], which can also be used as a steering mechanism for human-in-the-loop execution [45, 8]. Language-conditioned policies require special data and/or supervision to obtain and use the language labels, and it may be difficult to specify the exact steering needs through language alone. Other methods with fewer assumptions can learn goal-conditioned policies from trajectories by conditioning on learned latent embeddings [21, 28] or using future observations directly as the goal [41, 4, 38, 10, 2]. Like DynaGuide, these conditioned policies learn to accomplish the outcome represented by the goal observation. However, unlike DynaGuide, these approaches require training a policy that can take the goal condition as the input. DynaGuide's external dynamics model means that it works on top of any diffusion policy. It also supports multiple desirable and undesirable outcomes, whereas the observation-based goal conditioning typically supports only one desirable outcome. In our experiments, we compare the performance of DynaGuide with an observation goal-conditioned policy.

**Leveraging Dynamics Models to Improve Robot Policies.** Models can be trained to predict future states through latent dynamics by taking in an observation latent, action(s), and outputting a predicted future latent state [42, 31, 16, 14, 15, 56, 25]. These latent dynamics are commonly used for training reinforcement learning agents by simulating trajectories through the dynamics model [16, 14, 15, 40, 24] and have seen success on real robots [51]. They can also be used for Model Predictive Control (MPC) to generate trajectories directly [56, 43, 17, 13, 12, 47] and com-

|                               | Ours | PG | GC | GPC |
|-------------------------------|:----:|:--:|:--:|:---:|
| Untrained Goals (§4.2)        | ✓    | ✓  | ✗  | ✓   |
| Movable Objects (§4.1)        | ✓    | ✗  | ✓  | ✓   |
| Multiple Conditions (§4.3)    | ✓    | ✓  | ✗  | ✓   |
| Enhance Rare Behavior (§4.4)  | ✓    | ✓  | ✓  | ✗   |

Table 1: **Method Ability Comparisons.** Goal-conditioned (GC) and sampling methods (GPC) [39] can steer policy behavior, but GC is not prepared for untrained goals, and rare behaviors are challenging for GPC. External guidance methods address these shortcomings, and unlike Position Guidance (PG) [50], DynaGuide (Ours) can guide towards complicated objectives without precise coordinate input.

bined RL/Planning approaches [18, 19]. Especially relevant to DynaGuide are approaches that use dynamics models to filter samples from policies according to predicted value [34], VLM feedback [52], or engineered reward [39]. Like some of these prior works and especially Dino-WM [56], DynaGuide uses a transformer-based dynamics model to guide action creation. However, instead of sampling from the base policy, running RL, or MPC, DynaGuide steers the trained base policy directly through the action diffusion process.

**Inference-Time Steering.** After a policy has been trained, its behavior can still be changed without modifying its weights or requiring goal conditioning [48]. Safety value functions can intervene with a recovery policy or request human assistance when abnormal behavior is predicted [35, 27, 11]. Diffusion policies can be steered by skewing the initial noise distribution [49] or applying classifier-free guidance [20] that influences the action diffusion process [41] or the inverse-dynamics planning process [1]. While classifier-free guidance is effective for steering, it still requires training the policy on supplied conditions, akin to goal conditioning. The alternative steering method is *classifier guidance*, which leverages an external classifier to influence a diffusion generation process [9]. Classifier guidance in diffusion policies is akin to seeking an optimal trajectory defined by the classifier [22]. Applied to robot policies, classifier guidance can apply post-hoc constraints [32] or seek objects near a human-supplied keypoint [50]. DynaGuide is a classifier guidance approach, but it seeks to guide the policy with more complicated objectives using feedback from a trained dynamics model acting as the classifier. Dynamics-based diffusion guidance has succeeded in other applications like hardware generation [53] and DynaGuide brings it to robot policies.

## 3 DynaGuide Method

As shown in Fig. 2, we consider the problem of steering a diffusion policy $\pi_\theta(\mathbf{a}|o)$ during its inference process. We are given a set of **guidance conditions** $\mathcal{G} = \mathbf{g}^+ \cup \mathbf{g}^-$, which are represented as image observations that contains partial state information. Some guidance conditions show desirable outcomes $\mathbf{g}^+ = \{g_1^+, ...g_j^-\}$ and others are undesirable outcomes $\mathbf{g}^- = \{g_1^-, ...g_i^-\}$. Using $\mathcal{G}$, we want to create a $\pi_\theta'(a|o)$ such that for every $\mathbf{a} \sim \pi_\theta'(\cdot|o_t)$, the probability $p(\mathbf{g}^+|o_t, \mathbf{a})$ is as high as possible and the probability $p(\mathbf{g}^-|o_t, \mathbf{a})$ is as low as possible. To accomplish this, we train a

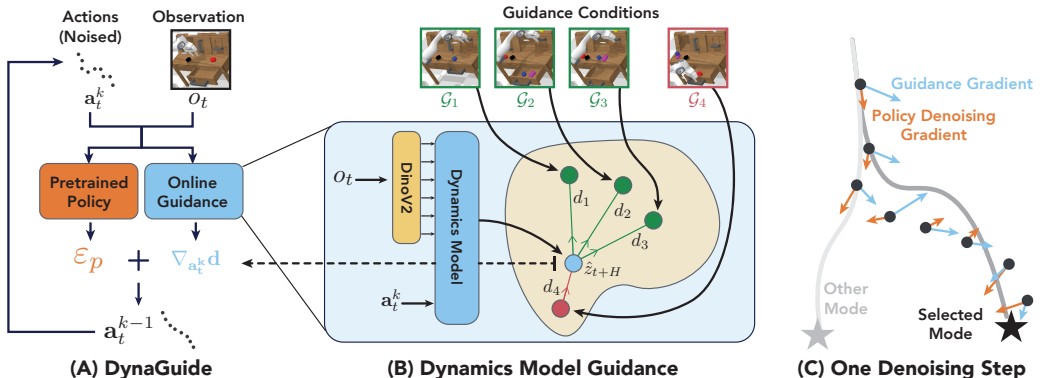

**(A) DynaGuide** **(B) Dynamics Model Guidance** **(C) One Denoising Step**

Figure 2: **Achieving Dynamics Guidance**. **(A)**: DynaGuide combines action denoising gradients $\varepsilon_p$ from the pretrained policy with a guidance gradient $\nabla_{a_t^k}\mathbf{d}$ that increases the likelihood of accomplishing a set of guidance conditions $\mathcal{G}$. **(B)**: Inside the guidance module, a dynamics model predicts future outcomes $\hat{z}_{t+H}$ and compares them to guidance conditions $\mathcal{G}$ (desired / undesired outcomes). We use the latent distances $d$ to define a guidance metric $\mathbf{d}$ (Equation 2) and take the gradient to get the guidance signal $\nabla_{a_t^k}\mathbf{d}$ used by DynaGuide. **(C)**: An example of one denoising step. The pretrained policy seeks behavior modes in the data, while the guidance gradient selects a particular mode.

dynamics model and use the model to approximate these probabilities (§3.1) and then use the gradient of that dynamics model to create $\pi'$ by modifying its action denoising process (§3.2).

## 3.1 A Dynamics Model Capable of Guidance

Creating a model to approximate $p(\mathbf{g}^+|o_t, \mathbf{a})$ and $p(\mathbf{g}^-|o_t, \mathbf{a})$ has two requirements: 1) the model needs to predict future outcome $\hat{o}_{t+H}$ from current observation $o_t$ and action sequence $\mathbf{a}$, 2) we need to compare $o_{t+H}$ to to the guidance conditions. Here, $H$ can be a large value or $t + H$ can be the end of the trajectory.

Past works on predicting future observations for planning and reinforcement learning have used *latent dynamics models* to satisfy these two requirements. These dynamics models $h_\theta(\phi(o_t), \mathbf{a})$ operate in a learned latent space $z_t = \phi(o_t)$. The recent work of Dino-WM [56] demonstrated the usefulness of latent distances in the pretrained DinoV2 image embeddings [37] for planning. Taking inspiration from Dino-WM, we leverage the same DinoV2 image latent space for our observation comparisons. Concretely, we use the patch embeddings from DinoV2 as $z_t = \phi(o_t)$ and train a transformer $h_\theta(\mathbf{a}, z_t)$ to predict a latent outcome representation $\hat{z}_{t+H}$. Because $\phi$ is frozen and we have access to full trajectories of data during training, it is sufficient to train $h_\theta$ through a regression objective (Eq. 1). For more details, refer to Appendix B.

$$\mathcal{L}(o_t, o_{t+H}, \mathbf{a}) = ||\phi(o_{t+H}) - h_\theta(\phi(o_t), \mathbf{a})||_2^2 \qquad (1)$$

With this dynamics model trained, we define a *guidance metric* that compares the predicted outcome with the guidance conditions. Given one guidance condition $g_i^+$, we want to approximate $p(g_i^+|o_t, a)$, which can be done by comparing the predicted outcome $\hat{z}_{t+H}$ to $g_i^+$. The condition $g_i^+$ is a visual observation, so we project it into the same latent space $z_i^+ = \phi(g_i^+)$. This allows us to directly compare $z_i^+$ and $\hat{z}_{t+H}$. If we roughly assume that the latent space is gaussian in structure, a reasonable approximation of $\log p(g_i^+|o_t, a)$ is proportional to $-||\hat{z}_{t+H} - z_i^+||_2^2$, a metric found in similar forms elsewhere for latent reinforcement learning [54, 33] and planning [56].

---

**Algorithm 1** DynaGuide (Inference-Time)

1: **Input**: Guidance Conditions $\mathbf{g}^+, \mathbf{g}^-$, Dynamics model $h_\theta$
2: **Input**: Action denoiser $\epsilon_\phi(a, o)$, current obs $o_t$
3: $\mathbf{a}^K \leftarrow$ Sample from $\mathcal{N}(0, I)$
4: **for** $k$ in $K$ to 1 **do**      ▷ Action Denoising
5:     **for** $i$ in 1 to $M$ **do**    ▷ Stochastic Sampling
6:         $\epsilon \leftarrow \epsilon_\phi(\mathbf{a}^k, o_t)$
7:         $\mathbf{d} \leftarrow$ Eq. 2
8:         $\hat{\epsilon}(\mathbf{a}^k, o_t) \leftarrow \epsilon - s\sqrt{1 - \bar{\alpha}_k}\nabla_{\mathbf{a}^k}\mathbf{d}$
9:         **if** $i < M$ **then**
10:             $\mathbf{a}^k \leftarrow$ Denoise $\mathbf{a}^k$ using $\hat{\epsilon}$
11:         **else**
12:             $\mathbf{a}^{k-1} \leftarrow$ Denoise $\mathbf{a}^k$ using $\hat{\epsilon}$
13:         **end if**
14:     **end for**
15: **end for**

When we try to maximize $\log p(\mathbf{g}^+|o_t, a)$ across the set of guidance conditions $\mathbf{g}^+$, we can try to maximize the combined probability of achieving each outcome $\log \sum_i p(g_i^+|o_t, \mathbf{a})$. The negative is true for the undesired outcomes. We use our approximation of $\log p(g_i^+|o_t, \mathbf{a})$ and a variance hyperparameter $\sigma$ to get the guidance metric $\mathbf{d}(\mathbf{g}^+, \mathbf{g}^-, o_t, \mathbf{a})$ (Eq. 2). The log-sum-exp acts as a soft maximum, a feature useful for diverse objectives (§4.3) and lower quality guidance conditions (§4.2). For a more detailed derivation of $\mathbf{d}$ and explanation of design choices, refer to Appendix B.3.

$$\mathbf{d} = \log \left[ \sum_i \exp \frac{-||\phi(g_i^+) - h_\theta(\phi(o_t), \mathbf{a})||_2^2}{\sigma} \right] - \log \left[ \sum_j \exp \frac{-||\phi(g_j^-) - h_\theta(\phi(o_t), \mathbf{a})||_2^2}{\sigma} \right] \quad (2)$$

We train the dynamics model using diverse robot interaction data. For the Calvin experiments §4.1-4.4 we use existing human-collected play data from the benchmark. For the real-world experiments §4.5, we use open-source data collected on the UMI interface [6]. Because we will be querying the dynamics model with noisy actions during the guidance process, we augment the training by adding gaussian noise to the actions using the same noise scheduler used during inference time, with a noise step picked from a geometric distribution. For more details about implementation, see Appendix B.

### 3.2 Guiding the Action Diffusion Process

To sample an action from a diffusion policy $\pi(\cdot|o_t)$, the policy predicts a sequence of denoising steps $\epsilon(\mathbf{a}^k, o_t)$, that moves an initial noise sample $\mathbf{a}^K \sim \mathcal{N}$ to a denoised action sequence $\mathbf{a}^0$. Diffusion models can be guided by a classifier $p(y|a)$ towards a particular set of samples that maximize $p(y|a)$ [9]. The guidance signal comes from the gradient of the classifier $\nabla_a \log p(y|a)$ and is combined with $\epsilon(\mathbf{a}^k, o_t)$ during the denoising process.

In our case, $y$ is the guidance conditions, and the $\log p(y|a)$ is our derived metric $\mathbf{d}$ in Eq. 2. Like past work [53, 41], we use the Denoising Diffusion Implicit Models (DDIMs) [46] as our sampling method. Under DDIM, the combined denoising and guidance signal is

$$\hat{\epsilon}(\mathbf{a}^k, o_t) = \epsilon(\mathbf{a}^k, o_t) - s\sqrt{1 - \bar{\alpha}_k}\nabla_{\mathbf{a}^k}\mathbf{d}(\mathbf{g}^+, \mathbf{g}^-, o_t, \mathbf{a}^k) \quad (3)$$

where $\bar{\alpha}_k = \prod_l^k \alpha_l$, $\alpha_l$ is the noise scheduler, and $s$ is a guidance scaling parameter [9]. The higher the $s$, the stronger the signal to adhere to the guidance requirements. However, past work has discovered that higher $s$ also create less smooth or even incoherent trajectories [50], creating a delicate balance between trajectory validity and adherence to guidance conditions. This phenomenon happens because $\nabla_a d$ can pull $\epsilon$ out of distribution, leading to erroneous noise predictions that compound as the denoising process continues. To resolve this, Wang et al. [50] proposed the *Stochastic Sampling* solution of running each denoising step $k$ multiple ($M$) times to stabilize the guidance signal's influence on the action denoising process through MCMC sampling. Empirically, this trick allows $s$ to be pushed higher to increase guidance success without sacrificing trajectory validity. See Algorithm 1 for an overview of DynaGuide. We investigate these hyperparameters in Appendix A.4.

## 4 Experiments

To understand the features, benefits, and limitations of DynaGuide, we conduct five sets of experiments on the Calvin environment (Figure 3, §4.1-4.4). To demonstrate its practical feasibility on real robots, we steer a publicly available robot policy in a real task using DynaGuide (§4.5). We also conduct experiments on a toy 2D navigation environment, which can be found in Appendix A.1. Wherever relevant, we compare DynaGuide to a representative set of baselines.

- **Base Policy.** This is the diffusion policy [7] that we steer with DynaGuide. This policy offers a fairly uniform distribution across valid behaviors.
- **DynaGuide-Sampling (GPC) [39].** Instead of diffusion guidance, we sample from the base policy multiple times and pick the action sequence that best satisfies metric $\mathbf{d}$ (Eq 2), an idea demonstrated in GPC-Rank with engineered reward functions [39]. We re-implement this idea with our dynamics model and use this baseline to explicitly test the benefits of diffusion guidance.

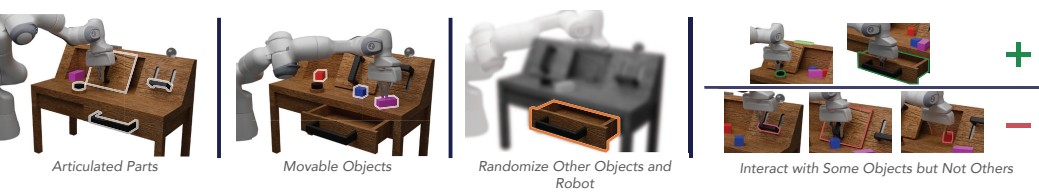

| Articulated Parts | Movable Objects | Randomize Other Objects and Robot | Interact with Some Objects but Not Others |
|---|---|---|---|
| **Fully-Specified Objective** | | **Under-Specified Objective** | **Multiple Objectives** |

Figure 3: **Experiment Setup**. In the CALVIN simulator [30], we propose four experimental setups designed to showcase DynaGuide and its advantages over other steering approaches. First, we test performance with high quality outcome observations as guidance conditions (*Fully-Specified Objective*). Next, we reduce the guidance condition quality by randomizing robot states and other states not relevant to the target object (*Underspecified Objective*). Finally, we look at how we can guide the base policy in complex ways, including achieving multiple behaviors and avoiding behaviors (*Multiple Objectives*).

- **Goal Conditioning.** We train a policy to take an additional visual observation as the goal. Past works in visual goal conditioning have used future observations as goals [2, 36, 41]. In our baseline, we take the common approach of using the last observation of each trajectory as the goal. We compare this model to DynaGuide by sampling from $\mathbf{g}^+$ as our goal during inference-time. We also test an intermediate goal-conditioning baseline in Appendix A.2.
- **Position Guidance (ITPS) [50].** Past work has achieved diffusion policy guidance that steers the robot to a particular location in 3D space supplied by a human. For objects that do not move, we replicate this guidance by finding the average location of the robot after accomplishing the desired behavior and using position guidance on that location.

## 4.1 Experiments 1 & 2: Steering in Complex 3D Environments

Our first two experiments examine how DynaGuide can steer robots to particular behaviors in the CALVIN environment [30]. The CALVIN setup allows a robot to interact with desks with drawers, switches, buttons, and cabinets (Fig. 3). It can articulate these objects and also rearrange a set of three colored cubes that are placed randomly on the table. For these two experiments, we focus on Guidance Conditions $\mathbf{g}^+$ that represent desired outcomes, taken as the last observation of demonstration trajectories showing the target behavior. We use these same observations for the goal-conditioned baseline by sampling $g_i^+ \in \mathbf{g}^+$ for each rollout as the goal.

In this set of experiments, we are interested in the ability of DynaGuide to steer the base policy towards behaviors that interact with the articulated table elements (**ArticulatedParts**) and randomized cube objects (**MovableObjects**). Overall, for every target behavior across both experiments, DynaGuide significantly increases the frequency of the target behavior over that of the base policy (Fig. 4, Top, Bottom-Left).

On **ArticulatedParts**, DynaGuide boosts the base policy's behavior by 8.7x and achieves an average target behavior success of 70%. Because the objects in **ArticulatedParts** stay in place, we can run the ITPS baseline [50]. Using position guidance boosts the target behavior over the base policy, but the reasoning abilities of the dynamics model allow DynaGuide to outperform it across the tasks (Fig. 4, Top). The location consistency of **ArticulatedParts** also means that the $\mathbf{g}^+$ outcomes closely match the true outcome of the current environment if the desired behavior is executed. Therefore, the goal-conditioned baseline is in-distribution and performs near perfectly. This is expected, as the goal-conditioned model was trained for this exact setup.

The **MovableObjects** experiment poses a new set of challenges. Instead of being fixed in location, the cube objects are small and randomly arranged every environment reset. All steering performances decrease in **MovableObjects**, but two notable differences arise. First, the GPC baseline performance drops to the base policy. Sampling approaches like GPC rely on actions drawn from the base policy. If these actions have high variance, then the selected actions may also have high variance. Compared to the steady guidance of DynaGuide, the increased variance leads to more erratic actions and impacts performance on precision tasks. The second difference is the drastic drop in goal conditioned performance. Unlike **ArticulatedParts**, the randomized cubes mean that not all outcomes in $\mathbf{g}^+$ are attainable in the current setup. For example, some outcomes in $\mathbf{g}^+$ interact with the blue cube on the left side of the table, while a test-time setup might have the blue cube on the *right* side. This

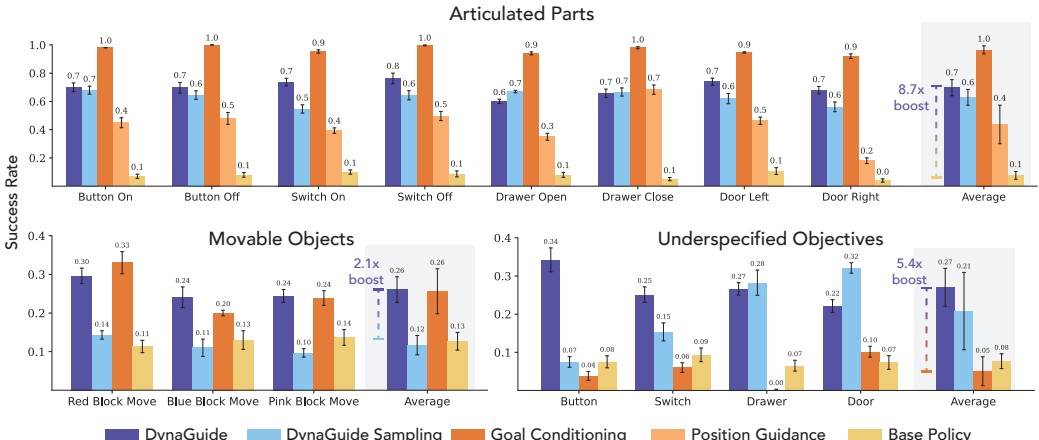

Figure 4: **Steering Ability and Robustness in the Calvin Environment** DynaGuide enhances the target behavior (horizontal axis) significantly across all experiments (Section 4.1). The goal conditioning baseline performs very well on a clean fixed articulated setup, but it drops steeply with lower goal quality while DynaGuide remains more robust (Section 4.2). For more precise tasks with movable cube objects, the active guidance in DynaGuide outperforms a sampling-based approach with the same dynamics model (Section 4.1)

mismatch means that the goal conditioned policy is out of distribution and is affected more than DynaGuide. We explore this phenomenon in the next experiment.

## 4.2 Experiment 3: Robustness to Lower Quality Guidance Conditions

The cube randomization in **MovableObjects** created a mismatch between *desired* outcome observation and *attainable* outcome in the current environment, a challenge that disproportionally impacted goal conditioning over DynaGuide. While **MovableObjects** had an inevitable mismatch, we explore a *deliberate* mismatch in this experiment that models a more practical use case of inference-time steering.

The two prior experiments had used $\mathbf{g}^+$ drawn from trajectories where the robot achieves the desired behavior. For each guidance condition, the combined robot's position and the target object's position fully indicate the desired behavior. For practical deployment, it is easier to take pictures of the desired scene *without* collecting trajectory demonstrations or placing the robot in the correct position. We mimic this requirement in **UnderspecifiedObjectives**, where we set the target articulated table part to the desired pose and randomize all other objects, including the robot. We use images from this setup as $\mathbf{g}^+$.

As expected, the lower quality guidance conditions pushed the goal-conditioned policy out of distribution, leading to average success rates below 10% (Fig. 4, bottom right). In contrast, the dynamics model guidance approaches (DynaGuide and DynaGuide-Sampling) were still able to increase the target behavior over base policy rates and outperformed the goal conditioned policy by 5.4x on average. DynaGuide also performed more consistently than its sampling counterpart, further illustrating the importance of active guidance. We attempt to strengthen the goal-conditioned baseline in Appendix A.2 by adding training data and implementing an intermediate goal-conditioning algorithm, but none of these approaches yielded consistent improvements over the tested goal-conditioned baseline here (Appendix A.2).

These under-specified objectives showcase the benefits of *separating* the policy and guidance. Although the guidance conditions are still mismatched for DynaGuide, the inputs $(\mathbf{g}^+, o_t, a)$ were all individually in-distribution to the parameterized encoder, base policy, and dynamics model of DynaGuide. The mismatch, therefore, occurs in the latent space of a large visual encoder. With nothing out-of-distribution, we face an easier challenge of extracting a meaningful signal from these latent distances. Fortunately, DynaGuide also makes use of *all* the conditions in the latent space for the guidance metric (Eq. 2). While each latent distance $d_i$ might be noisy and inaccurate, the combined metric will cancel some of the noise, leading to meaningful guidance.

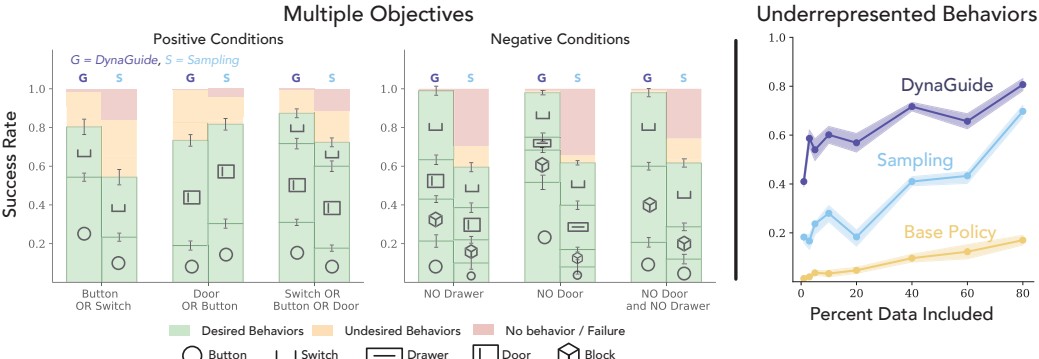

Figure 5: **Multiple Objectives and Underrepresented Behaviors.** DynaGuide is able to steer the base policy towards multiple behaviors while minimizing other behaviors and failures (Left). DynaGuide is also able to avoid undesired behaviors by performing other behaviors successfully (Middle). On these complicated objectives and in lower data regimes (right), DynaGuide performs better than sampling approaches.

## 4.3 Experiment 4: Steering with Multi-Objectives

Separating the guidance and the policy improves the robustness of DynaGuide to lower quality guidance conditions, and it also allows the guidance signals to be *combined*, steering the policy towards a multimodal set of desired outcomes—and even *away* from a set of negative outcomes. This capacity is not possible for an unmodified goal-conditioned policy that takes a single observation as the goal.

In **MultiObjectives**, we add multiple target behaviors to $\mathbf{g}^+$ and populate $\mathbf{g}^-$ with undesired behaviors. Good steering will allow all represented behaviors in $\mathbf{g}^+$ to be executed while minimizing behaviors in $\mathbf{g}^-$ and overall behavior failure rate (e.g., doing nothing). Indeed, DynaGuide steers the policy towards multiple target behaviors with fair representation of each behavior (Fig. 5, Left), achieving up an average of 80% success with nearly no full behavior failures. When faced with avoiding behaviors in $\mathbf{g}^-$, DynaGuide achieves nearly perfect success in avoiding the undesired behavior while executing a variety of other behaviors (Fig. 5, Middle). For an additional comparison against Classifier-Free Guidance (CFG) [20], refer to Appendix A.5.

Because DynaGuide and GPC both use the same dynamics model for steering, they can both steer to multiple objectives. However, the sampling approach of GPC had lower overall success of desired behaviors and allowed more execution of undesired behaviors, especially when trying to avoid outcomes in $\mathbf{g}^-$ (Fig. 5, Middle). When avoiding $\mathbf{g}^-$, GPC also produces more behavior failures where nothing is executed during the trajectory horizon. Sampling approaches like GPC rely on the base policy to readily produce actions that satisfy the objective. However, especially when the robot is close to an object, the base policy might only sample actions for one behavior, making this behavior impossible to avoid. In contrast, active guidance during the diffusion process can seek rare modes in the action distribution, leading to a higher chance of accomplishing complicated objectives like the ones in this experiment.

## 4.4 Experiment 5: Enhancing Underrepresented Behaviors

In **MultiObjectives**, we observed that the GPC sampling approach struggles to satisfy complicated objectives because it relies on adequate samples from the base policy. In **UnderrepresentedBehaviors**, we test this feature explicitly. Trained robot policies will offer behaviors based on its representation in the training data. However, when we steer a policy, we also want to steer it into behavior modes that are *underrepresented* in the training data.

To simulate data underrepresentation, we reduce the presence of Switch-On behavior in the training set of the base policy. As expected, the base policy offers more Switch-On behavior as more data is included (Fig. 5, Right). GPC also increases with more data representation, as it can boost the behavior by selecting the right action. Most importantly, DynaGuide consistently outperforms its sampling counterpart at these lower data regimes, achieving 40% success rate with only 1% of the original Switch-On behavior included in the trained base policy (Fig. 5 Right). By directly influencing the

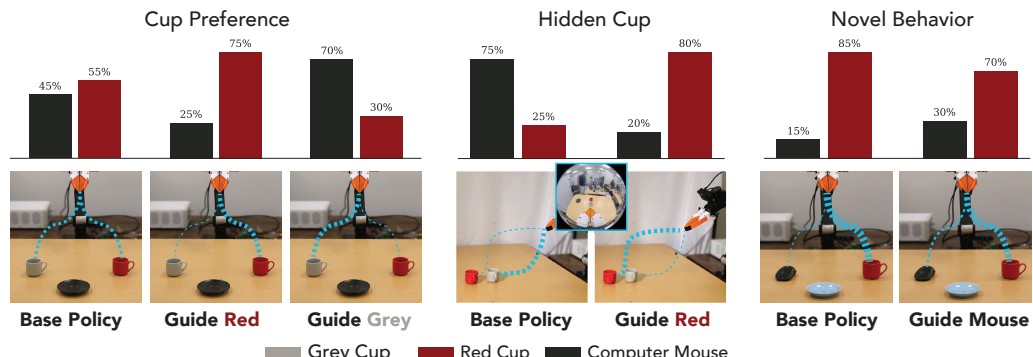

Figure 6: **Real World Experiments** On a cup arrangement task (Left, Center), we show that DynaGuide can guide a pretrained robot policy to express preference over cup color. Using the same policy, we also show that we can create novel mouse-grabbing behavior (Right) by leveraging the additional knowledge inside the trained dynamics model.

action diffusion process, DynaGuide can leverage the knowledge of the dynamics model to seek areas in the action distribution that satisfy an objective, even if they are underrepresented.

## 4.5   Experiment 6: Steering Off-The-Shelf Policies with a Real Robot

In the simulated Calvin environment, we used existing data to train both the base policy and the dynamics model. DynaGuide works with this setup, but it should also work with *any pretrained diffusion policy* and it should also work on *real robots*. To test these two claims, we set up a real robot experiment leveraging a publicly available pretrained policy [6] that picks up a cup and places it on a saucer. We train the dynamics model using open-source data from the UMI collection interface [6] and a set of demonstrations collected on the experimental setup. For more details about the setup, refer to Appendix B.5.

We conducted three experiments on this real-world setup. The first two, **CupPreference** and **HiddenCup**, present the robot with two different-colored cups equidistant from the robot's starting gripper. The base policy will select a cup at random and place it on the saucer. Applying DynaGuide in **CupPreference** creates a preference for a cup color (Fig. 6 Left), leading to an average target behavior success of 72.5%. In **HiddenCup**, the red mug is hidden behind the grey mug. The base policy will typically go for the closer grey mug, but the active guidance in DynaGuide enables the robot to find the red cup 80% of the time (Fig. 6 Middle).

Inspired by the performance of DynaGuide with data representation at 1% (§4.4), we tested the ability of DynaGuide to steer the pretrained policy towards a novel behavior: touching a computer mouse. In **NovelBehavior**, the dynamics model was trained on various object manipulations, including mugs and computer mice from other open-source datasets [26]. We used the same off-the-shelf base policy that was only trained to arrange mugs. Although the steered policy still expressed preference for the mug, DynaGuide was able to *double* the number of interactions with the novel object.

## 5   Conclusion and Discussion

In this paper, we proposed a novel method of steering pretrained diffusion policies by using a separately trained latent dynamics model. We demonstrated the ability of DynaGuide to enhance target behavior performance across simulated and real experiments, outperforming baselines like position guidance and goal conditioning in some or all setups. Finally, we demonstrated that DynaGuide can be steered to achieve multiple behaviors, avoid undesirable behaviors, and enhance behaviors that were underrepresented in the training of the base robot policy. Real-world robot deployment requires robots to be highly steerable in their skills, and DynaGuide shows one potential avenue for this ability.

### 5.1   Limitations and Future Work

DynaGuide can successfully steer pretrained policies towards particular objectives, but currently it is difficult to specify the *method* of achieving this objective. This limitation is due to the current form of

guidance conditions as observations that represent desired/undesired *outcomes*. There are other, more complicated guidance modalities, including language and kinesthetic demonstrations. Future work includes enabling multi-modal guidance conditions, which can hopefully enable very fine-grained guidance for all parts of the trajectory. In the future, we also hope to add an ability for the base policy to remember past guidance, consolidate the knowledge, and apply it automatically to new tasks.

## 5.2 Acknowledgments

Maximilian Du is supported by the Knight-Hennessy Fellowship and the NSF Graduate Research Fellowships Program (GRFP). This work was supported in part by the NSF Award #2143601, #2037101, and #2132519, and Toyota Research Institute. We would like to thank ARX for the robot hardware and Yihuai Gao for assisting with the policy deployment on the ARX arm. We appreciate Zhanyi Sun for discussions on classifier guidance and all members of the REAL lab at Stanford for their detailed feedback on paper drafts and experiment directions. The views and conclusions contained herein are those of the authors and should not be interpreted as necessarily representing the official policies, either expressed or implied, of the sponsors.

The Calvin Experiments (Sections 4.1 - 4.4) were made possible with the Calvin benchmark codebase [30]. The diffusion policy was adapted from the Robomimic repository [29]. The dynamics model was inspired by the Dino-WM implementation [56] and leverages representations from Dino-V2 [37].

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

# A Additional Experiments, Visualizations, and Ablations

## A.1 Steering in a Simple 2D Block Task

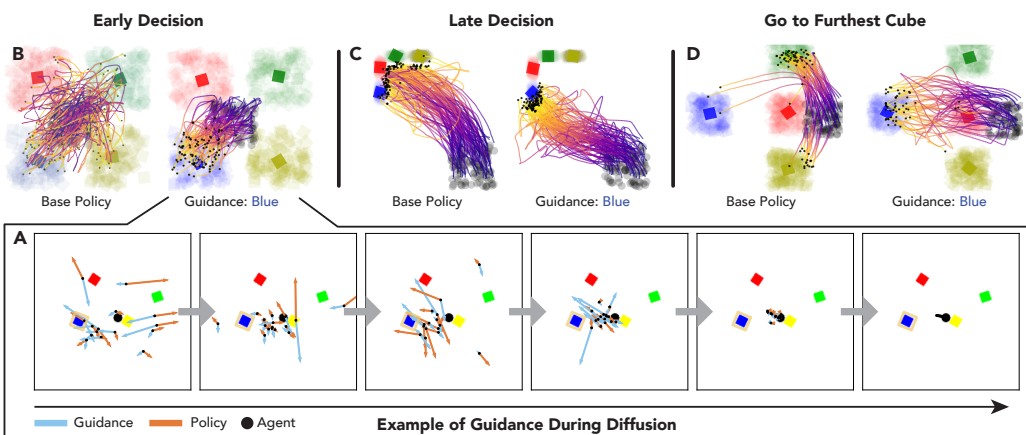

Figure 7: **The BlockTouch Environment**. This 2D environment requires the agent (black circle) to navigate to a colored target (square). The base policy picks an arbitrary target, and DynaGuide steers it towards a particular color (**A**). This steering works to express square color preference (**B**). The dynamics model also enables the base policy to accomplish novel tasks not explicitly trained on the base policy, including picking a target from a tight cluster of squares (**C**) and navigating past three close cubes to a far target (**D**). In visuals **B - D**, the distribution of squares are indicated by the shaded regions. The average square location is represented by the solid square.

As an additional way of understanding how guidance works in DynaGuide, we look at a toy **Block-Touch** environment. In **BlockTouch**, the agent navigates a 2D environment and touches a colored square. We train the vision-based base policy and dynamics model on synthetic data that shows navigation to squares in any location of any color. Then, during test time, we use the dynamics model to guide the policy towards cubes of one particular color. In Figure 7A, we visualize the creation of an action chunk under dynamics guidance. In the initial noisier steps, the guidance signal and the denoising policy signal compete for the action's direction. In the example, the agent is nearly touching the yellow square, so the base policy wants to denoise the action towards the yellow square. The guidance wants to move the agent towards the blue square. The vector sum of these two forces pushes the actions towards the blue square. In the later stages of denoising, the guidance and the policy signals start to work together to craft the final action sequence that points to the blue square. This guidance works in situations where early decisions are important for hitting the correct color (Figure 7B) and in situations where late decisions are important (Figure 7C).

Even though the dynamics and base policies are both trained on random navigation behavior, we can leverage the dynamics model to accomplish novel tasks not explicitly present in the base policy. The *Late Decision* setup (Figure 7C) contains squares that are closer together than the training data, requiring high navigational precision. The *Go to Furthest Cube* setup (Figure 7D) places the target much further away than three squares, requiring deliberate navigation around the closer squares. The base policy goes to the closer squares, but the dynamics model was able

|  | **Red** | **Grey** |
|---|---|---|
| DynaGuide | **75%** | 25% |
| DynaGuide-Sampling | 60% | 40% |
| Base Policy | 55% | 45% |

Table 2: **CupPreference Additional Result** We add a baseline to the GuideRed condition of CupPreference (Section 4.5. While DynaGuide-Sampling improves over baseline, DynaGuide achieves the best guidance.

to reliably guide the policy between the squares and to the target. Note that the visuals in **B-D** show square distributions and the black dot at the yellow end of the trajectory shows the first and only contact with a square in the environment.

## A.2 Additional Goal-Conditioning Baselines

In Experiments 1-3, we use a simple goal conditioning baseline by using the last trajectory state as the goal representation. To maintain fairness with the base policy, we also train the goal conditioned model on only the test environment's data in Calvin. However, we note that the dynamics model in

DynaGuide is trained with multi-environment data. To maintain fairness with the total data exposed to all parts of a method, we train another goal-conditioned policy on the same multi-environment data. When evaluated on **UnderspecifiedObjectives**, this additional data improved performance across some steering targets, although DynaGuide still outperformed this augmented goal conditioning baseline (GC w/ All Data, Table 3).

Finally, we try to apply an algorithmic improvement on the goal conditioning baseline by implementing Hindsight Experience Replay [2]. Instead of conditioning the policy on only end states during training, we also sample intermediate future states to broaden the goal distribution. We find that this improves steering success across some tasks but not others (GC w/ HER, Table 3). Overall, we find that goal conditioning can be improved for select tasks with additional data or algorithmic changes, but these augmentations do not raise the performance of this baseline to that of DynaGuide.

|  | button_on | button_off | switch_on | switch_off | drawer_open | drawer_close | door_left | door_right |
|---|---|---|---|---|---|---|---|---|
| DynaGuide | 0.33±0.031 | 0.36±0.032 | 0.24±0.017 | 0.26±0.024 | 0.26±0.015 | 0.27±0.017 | 0.25±0.018 | 0.19±0.015 |
| Original GC | 0.023±0.0080 | 0.053±0.015 | 0.073±0.018 | 0.047±0.0067 | 0.00±0.00 | 0.0033±0.0033 | 0.12±0.012 | 0.080±0.016 |
| GC w/ All Data | 0.054±0.017 | 0.10±0.012 | 0.064±0.015 | 0.047±0.015 | 0.00±0.00 | 0.00±0.00 | 0.095±0.026 | 0.075±0.021 |
| GC w/ HER | 0.0067±0.0042 | 0.087±0.030 | 0.027±0.011 | 0.020±0.0073 | 0.00±0.00 | 0.00±0.00 | 0.21±0.023 | 0.13±0.021 |

Table 3: **Stronger Goal Conditioned Methods.** Strengthening the goal conditioning baselines with additional data or hindsight replay [2] shows some improvement in steering ability, but as tested on **UnderspecifiedObjectives** (Exp 3), they still perform worse than DynaGuide.

## A.3 Additional Real Robot Baseline

The real robot experiments (Section 4.5) demonstrated the ability for DynaGuide to steer pretrained policies. To further demonstrate the performance of DynaGuide on real robots, we add DynaGuide-Sampling baseline for one real robot task. As seen in Table 2, influencing the action sampling with the diffusion model leads to an improved success rate of the target task over baseline, but the classifier guidance of DynaGuide still yields a higher success rate of the target task.

## A.4 Ablations and Hyperparameter Investigation

To understand how various hyperparameters and components of DynaGuide contribute to the final success rate, we run a set of experiments on the `Switch-On` task in the **ArticulatedParts** experiment. In general, we observe that the two main hyperparameters $\sigma$ (Eq. 2) and $s$ (Eq. 3) are robust to reasonable changes, meaning that DynaGuide is not difficult to tune. At very low values of $s$, success rates are lower because there is not enough guidance. Success rates increase with higher strength until the strength becomes too high and creates instability. Very low values of $\sigma$ decreases guidance success by making the guidance conditions too far away for any meaningful signal. Increasing the number of stochastic sampling steps $M$ generally increases the performance of guidance at a cost of computational expense. Surprisingly, DynaGuide is very robust to the number of guidance conditions $\mathcal{G}$, achieving comparable guidance even with one guidance condition.

In practice, we use a guidance strength and $\sigma$ chosen from a gridsearch for each experiment. These parameters are often very similar across tasks in the same environment (Table 4), which means that switching target tasks will not require extensive hyperparameter search. We use stochastic sampling $M = 4$ for our experiments as a balance of stability and computation efficiency. We use 20 guidance conditions per task, which is more than sufficient for `Switch-On` but is more important for underspecified goals (§4.2) and harder objectives.

Critically, we discover that pretraining the dynamics model with a noised action is essential for performance (Fig. 9). This makes sense as the noise would otherwise be out of distribution.

In experiments 1-5, we leverage a more diverse dataset collected across four different Calvin environments to train the dynamics model. This data magnitude improves training stability and reduces overfitting. However, because we test only on a single Calvin environment, we conduct a

| Task | Scale | $\sigma$ |
|---|---|---|
| switch_on | 1.5 | 30 |
| switch_off | 1.5 | 30 |
| drawer_open | 1.0 | 40 |
| drawer_close | 1.0 | 40 |
| button_on | 1.0 | 30 |
| button_off | 1.0 | 30 |
| door_left | 1.5 | 30 |
| door_right | 2.0 | 15 |

Table 4: **Sampling Hyperparameters per Task.** Per environment, the optimal parameters are very similar across tasks.

data ablation and train the dynamics model only on data from that environment. As seen in Table 5, reducing the dynamics training data also reduces the steering success, although the performance is still above baseline (Experiment 4.1). This ablation demonstrates that DynaGuide can transfer information from related environments to help steer a policy in a target environment, thereby reducing the data requirement for training a dynamics model in a new environment.

| | button_on | button_off | switch_on | switch_off | drawer_open | drawer_close | door_left | door_right |
|---|---|---|---|---|---|---|---|---|
| All Environments | 0.70±0.031 | 0.70±0.038 | 0.74±0.027 | 0.76±0.037 | 0.60±0.015 | 0.66±0.029 | 0.74±0.025 | 0.68±0.026 |
| Target Environment Only | 0.66±0.023 | 0.69±0.038 | 0.54±0.043 | 0.58±0.031 | 0.51±0.022 | 0.53±0.024 | 0.65±0.019 | 0.57±0.036 |

Table 5: **Reducing Training Data in Dynamics Model.** Ablation result from **ArticulatedParts** (Section 4.1) using DynaGuide with a dynamics model trained only on the D-split of Calvin.

## A.5 Classifier-Free Guidance Baseline

An additional relevant baseline is the *Classifier Free Guidance* (CFG) [20] that leverages versions of the diffusion policy itself to guide the diffusion process. We implement and run CFG as a baseline for **MultiObjectives**. CFG can guide towards multiple objectives, a feat that the goal-conditioned baseline was not able to accomplish. Table 6 presents these additional results for a CFG baseline, which should be compared to those of DynaGuide and DynaGuide-Sampling in Figure 5. As expected, CFG can successfully guide the policy towards multiple objectives, including accomplishing multiple behaviors and avoiding other behaviors. However, the total success rate for positive guidance (first three rows in Table 6) is lower than that of DynaGuide (Figure 5).

The CFG is also able to steer the policy away from behaviors, as seen in the last three rows of Table 6. The success rates of the undesired behaviors for CFG and DynaGuide are within one standard deviation. However, while DynaGuide nearly always offers a valid alternative behavior (Figure 5), the CFG baseline fails to execute a behavior more than 10% of the time for all three tested conditions (No Behavior, Table 6). This is an expected result, as the log-sum-exp latent selection method of DynaGuide allows the guidance to find an appropriate behavior and minimize the influence of other contradictory behaviors. In contrast, the CFG implementation gives equal weight to all guidance signals. The lack of a latent distance makes it difficult to select the most relevant guidance. Contradictory guidance signals can influence the diffusion process negatively, leading to poorly-formed actions and unsuccessful behaviors.

| | Button | Switch | Drawer | Door | Blocks | No Behavior |
|---|---|---|---|---|---|---|
| Button or Switch | 0.18±0.01 | 0.37±0.02 | 0.05±0.02 | 0.05±0.00 | 0.29±0.02 | 0.05±0.01 |
| Button or Door | 0.37±0.01 | 0.09±0.01 | 0.01±0.00 | 0.28±0.03 | 0.23±0.02 | 0.02±0.01 |
| Switch or Button or Door | 0.17±0.02 | 0.29±0.04 | 0.03±0.00 | 0.23±0.02 | 0.25±0.01 | 0.03±0.01 |
| NO Drawer | 0.16±0.03 | 0.23±0.02 | 0.01±0.00 | 0.09±0.00 | 0.35±0.02 | 0.17±0.02 |
| NO Door | 0.23±0.03 | 0.15±0.02 | 0.12±0.04 | 0.01±0.00 | 0.38±0.04 | 0.11±0.02 |
| NO Drawer NO Door | 0.19±0.02 | 0.25±0.03 | 0.02±0.01 | 0.00±0.00 | 0.42±0.05 | 0.12±0.00 |

Table 6: **Additional Classifier-Free Guidance (CFG) Baseline**. Behavior distribution for a CFG baseline on the **MultiObjectives** Experiment shows that CFG can guide towards multiple objectives but struggles with avoiding specific behaviors without an increase in overall failures (lower three rows).

# B Implementation Details

## B.1 DynaGuide Implementation Details

**Base Diffusion Policy** The diffusion policy is trained to take a 2-step history stack of visual observations and robot proprioception and predict a chunk of 16 actions. It uses a Resnet-18 image encoder that conditions a U-Net with 4 encoding and 4 decoding layers. This U-Net predicts the noise during training using the standard diffusion policy objective [7]. During inference, we use the noise predictor with the DDIM noise scheduler to craft the generated action chunk. Together, the diffusion policy has around 18 million parameters. We use the Adam optimizer with a learning rate of 1e-4. We train the model for 200k gradient steps with a batch size of 16. All parts of the diffusion policy are trained

together using expert data. During execution, 14 actions are executed in the environment open-loop before the policy is queried again.

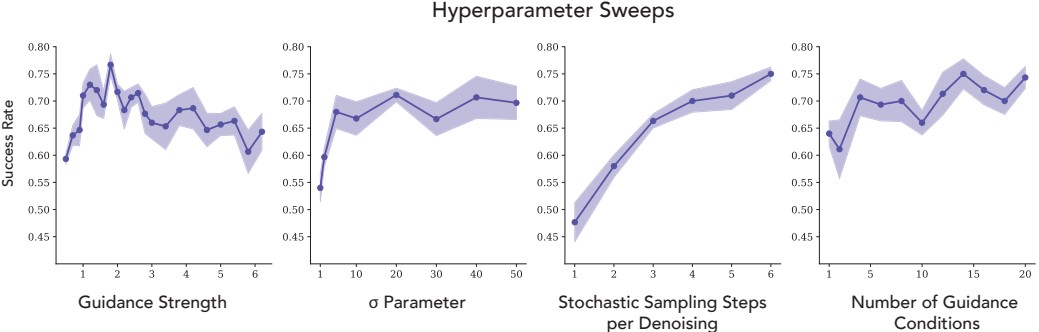

Figure 8: **Hyperparameter sweeps and ablations**. We look at the impact of inference-time hyperparameters and noise pretraining on final performance. For *guidance strength*, we hold $\sigma = 40, M = 4$, where $M$ is the number of stochastic sampling steps. For *$\sigma$ parameter*, we hold $s = 1.5, M = 4$ where $s$ is guidance strength. For *Stochastic Sampling*, we hold $s = 3, \sigma = 40$. We use a higher $s$ to demonstrate the impact of stochastic sampling on stability. For *Guidance Conditions* we use $s = 1.5, \sigma = 40, M = 4$.

**Dynamics Model**. We use the DinoV2 patch embeddings as the latent space, which provides 256 patches of embeddings with size 384. While this is a large embedding size ($256 \times 384$), past work has shown the effectiveness of this exact embedding in robot planning [56]. Empirically, both in past work and in our development, we discovered that the CLS token in DinoV2 ($1 \times 384$) does not carry as much spatial information needed for meaningful latent guidance. For the latent dynamics model, we use a 6-layer Transformer Encoder with 8 heads that takes in the 256 patches, a token for robot proprioception, and an action token. This action token is created by embedding each action in $\mathbf{a}^k$ using the same MLP and concatenating the embeddings into the token. We use a learned position embedding. The first 256 outputs of the transformer are taken as the predicted future state. We regress these patches to the DinoV2 embeddings of the last (or later) observation of the trajectory. For the CALVIN experiment, we regressed to the final latent of the trajectory. For the real-world experiment, we regressed to the latent of the observation after executing the proposed chunk of actions in the environment. We use the Adam optimizer at a learning rate of 1e-4. Because the DinoV2 embedding is frozen, the transformer can be trained end-to-end using a regression objective. This transformer has approximately 16 million parameters. We train the model for 600k gradient steps with batch size 16, a point past model convergence. During development, we showed that validation performance depends strongly on the action's presence, indicating that the dynamics model has learned to listen to actions.

To account for noisy actions during guidance, we add noise to the actions used to train the dynamics model. 50% of the fed actions are noiseless, and 50% are noised based on the same DDIM scheduler used during inference. We select the noising step using a geometric random variable with an expected noise step of 20. The scheduler starts at step 100 with pure gaussian noise, so at step 20, it is a noisy but still meaningful action that the dynamics model can use.

**Inference-Time Steering**. Before inference-time, we take the guidance conditions and pre-compute their embeddings. Then, during inference-time, we compute the objective (Eq. 2). Empirically, the Euclidean distance yields more stable results in lieu of squared distance in Eq. 2. We backpropagate this metric through the dynamics model and use the gradient with respect to the input actions as the guidance signal. Like the base policy, we use the DDIM sampler to reduce inference-time computation from 100 steps to 10 steps. To implement stochastic sampling, we repeat the same denoising step multiple times within the denoising loop.

**Compute Hardware**. All policies and dynamics models were trained on single RTX 3090 GPUs with 24GB VRAM, taking 24-48 hours to convergence. The dynamics model is $\approx$15M trainable paramters, which takes up $\approx$4GB of GPU memory during training and inference. All experiments were conducted on single RTX 3090 GPUs taking 10-20 minutes per seed per task. Development of the method and experiments were also conducted on RTX 3090 GPUs, with the total compute-hours for development estimated at 10-20 times the results shown in the paper. Most of this compute was spent developing the dynamics model.

## B.2 Implementation of Baselines

**Goal Conditioning**. We use the provided state-conditioning implementation in Robomimic [29] and condition the diffusion policy on an additional input consisting of the final observation in the trajectory. We train this policy using the same hyperparameters as the base diffusion policy.

**Position Guidance (ITPS) [50]**. The ITPS algorithm excels at steering the policy towards a target point in 3D specified by a human. To compare ITPS to our approach that does not require human specification, we manually compute the average final position of the robot for a set of 20 trajectories showing the desired behavior. We then use this position as the target point in ITPS.

**Sampling (GBC) [39]**. Previous work that introduced GBC-Rank used an objective function to rank action samples from a policy to select the best sample. We take that idea and apply it to our guidance metric (Eq. 2). We sample the base policy 5 times per inference and pick the best action to execute based on the metric.

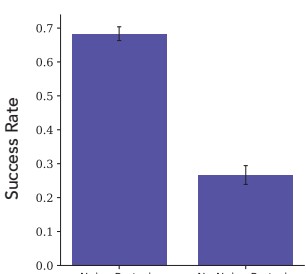

Figure 9: **Ablating Noise Pretraining**. Removing the dynamic model's exposure to noised actions greatly decreases its ability to steer the action diffusion process.

**Classifier-Free Guidance [20]**. We use the same goal-conditioned architecture as the goal-conditioned baseline. However, during training, we randomly zero the goal condition to represent the unconditional input needed in classifier-free guidance. During inference, we compute the classifier-free guidance noise estimation using the same conditions used in the goal-conditioned policy baseline.

## B.3 Distance Metric: Derivation and Design Choices

In this section, we will conduct a more detailed derivation of the metric shown in Eq. 2. While other works [56] simply assume the metric to be Euclidian, we use a probabilistic motivation to understand the log-sum-exp, which we discovered empirically to be very important for the performance of DynaGuide. For the first part of this derivation, we will only talk about the desired outcomes $\mathbf{z}^+$, and the undesired outcomes follow by negation.

We start from the very rough approximation that the latent space is Gaussian with diagonal variance $\Sigma$. It is an approximation that methods using squared latent distances already implicitly make. Here, the probability $p(z_i^+|\hat{z}_{t+H}) = \mathcal{N}(z_i^+, \mu = \hat{z}_{t+H}, \Sigma = \sigma I)$. The log probability can be computed as follows:

$$\log p(z_i^+|\hat{z}_{t+H}) = \log \frac{1}{(2\pi)^{n/2}|\sqrt{|\Sigma|}} \exp\left(-\frac{1}{2}(z_i^+ - \hat{z}_{t+H})^T \Sigma^{-1}(z_i^+ - \hat{z}_{t+H})\right)$$

$$= -\frac{1}{2}(z_i^+ - \hat{z}_{t+H})^T \Sigma^{-1}(z_i^+ - \hat{z}_{t+H}) + \underbrace{\log \frac{1}{(2\pi)^{n/2}|\sqrt{|\Sigma|}}}_{\text{Constant}}$$

$$= -\frac{1}{2\sigma}(z_i^+ - \hat{z}_{t+H})^T(z_i^+ - \hat{z}_{t+H}) + C$$

$$= -\frac{1}{2\sigma}||z_i^+ - \hat{z}_{t+H}||_2^2 + C$$

There are different ways of combining the influence of multiple $z_i^+$ in $\mathbf{z}^+$. One way that empirically *does not* work is adding the log-probabilities together:

$$d = \sum -\frac{1}{2\sigma}||z_i^+ - \hat{z}_{t+H}||_2^2$$

This is not a surprise, as adding log-probabilities is equivalent to *multiplying* the probabilities together. As previously discussed, not all the $z_i^+$'s are achievable and some may be mutually exclusive. For example, in the multi-objective setup (§4.3), some $z^+$ might correspond to pressing the button and

others to opening the drawer. The influence of any single $z_i^+$ is too powerful in this setup because of the product.

In contrast, it is more sensible to *add the probabilities*. This way, changes that generally increase the likelihood of the $z_i^+$'s will increase the metric, even if it decreases the likelihood of some other competing $z_i^+$'s. To add the probabilities, we can do the following:

$$
\begin{aligned}
\log p(\mathbf{z}^+|\hat{z}_{t+H}) &= \log \sum_i p(z_i^+|\hat{z}_{t+H}) \\
&= \log \sum_i \exp\left(-\frac{1}{2\sigma}||z_i^+ - \hat{z}_{t+H}||_2^2 + C\right) \\
&= \log \sum_i \exp(C) \exp\left(-\frac{1}{2\sigma}||z_i^+ - \hat{z}_{t+H}||_2^2\right) \\
&= C + \log \sum_i \exp\left(-\frac{1}{2\sigma}||z_i^+ - \hat{z}_{t+H}||_2^2\right)
\end{aligned}
$$

Which creates an intuitive result, as the log-sum-exp is a soft maximum. The metric focuses on the distances that are closer to the desired outcome. The value of $\sigma$ modulates the sharpness of the soft maximum.

We compute the same metric for $\mathbf{z}^-$. Because we want results in $\mathbf{z}^-$ to be mutually exclusive to $\mathbf{z}^+$, we want to *divide* the probabilities:

$$
d = \frac{p(\mathbf{z}^+)}{p(\mathbf{z}^-)}
$$

Intuitively, it means that there is a strong gradient incentive to push $p(z^-)$ as small as possible, which is not as strong if we were to use $p(\mathbf{z}^+) - p(\mathbf{z}^-)$. We observe the benefits of the division empirically. This setup corresponds to subtracting the undesirable outcome metric from the desirable outcome metric. To create the final $\mathbf{d}$, we absorb the constant 2 into $\sigma$ and discard the constant $C$. This gives us the expression in Eq. 2:

$$
\mathbf{d} = \log\left[\sum_i \exp \frac{-||\phi(g_i^+) - h_\theta(\phi(o_t), \mathbf{a})||_2^2}{\sigma}\right] - \log\left[\sum_j \exp \frac{-||\phi(g_j^-) - h_\theta(\phi(o_t), \mathbf{a})||_2^2}{\sigma}\right]
$$

Empirically, we discover that Eq. 2 works best with the squared L2 distance substituted with Euclidean distance. This deviation from the theoretical result is best attributed to the Gaussian latent assumption of DinoV2 not being fully true to the real embedding space.

In practice, $\mathbf{g}^+$ can be either a future state $(t + H)$ or a final state $(T)$. Using a future state $t + H$ is easier to train for the dynamics model but it is harder to find a correct $\mathbf{g}^+$ during inference-time guidance, as it must be a state achievable $H$ steps from the current timestep. Alternatively, predicting a final state $T$ means that final states from demonstration trajectories can be used as $\mathbf{g}^+$, making the choice of inference-time guidance examples straightforward. However, training a dynamics model to predict final state can be difficult for long-horizon tasks. We find that the Calvin environment enables end state prediction $(T)$, but the real world robot environment requires setting $H = 48$ steps into the future. We feed entire trajectories as $\mathbf{g}^+$ and rely on the log-sum-exp latent selection to dynamically pick the most relevant guidance points. The choice of $\mathbf{g}^+$ is an important design choice and it depends on the environment and the data.

### B.4 Experimental Setup: Simulation

**CALVIN Environment: Data**. Although CALVIN is used for benchmarking, we just use the CALVIN tasks for the data and our own sets of experiments. The CALVIN data is provided as a

continuous set of transitions. We used privileged state information to segment these transitions into trajectories showing one behavior per trajectory: switch on, switch off, drawer open, drawer close, door left, door right, button on, button off, red touch, red displace, blue touch, blue displace, pink touch, pink displace. We did our own segmentations to be consistent with our evaluation criteria (See later paragraphs). The segmentations were also important to extract end observations for the goal-conditioning baseline and training the dynamics model.

**CALVIN Environment: Base Policy**. We use the CALVIN-D dataset to train the base policy. We use third-person observation, wrist camera observation, and full robot proprioception for the base diffusion policy. Because the demonstrations show a large variety of behavior, the trained base policy also offers a wide variety of behaviors.

**CALVIN Environment: Dynamics Model**. Because the dynamics model can take a wider range of data, we train the dynamics model on the CALVIN-ABCD dataset, which is the full data split provided by the benchmark. We discover that adding the non-relevant environments improved convergence and reduced overfitting. We use the third-person observation and full robot proprioception as inputs to the dynamics model. The short tasks horizon of the CALVIN tasks meant that it was advantageous to train the dynamics model to predict the final state observation of the single task trajectory. We use the trajectory segmentations to obtain the final observation target during training.

**CALVIN Environment: Evaluations**. We segment the provided validation CALVIN dataset and randomly select 20 trajectories per desired behavior to extract $\mathbf{g}^+, \mathbf{g}^-$ using the last state observation (except for the **UnderspecifiedObjectives** experiment). For each evaluation, we perform 50 trials with a horizon of 400. We use privileged state information to monitor the object interactions and we stop the rollout when an object is sufficiently articulated or moved. If no object is sufficiently moved or articulated after 400 steps, we count the trial as a failure (no behavior). For each target behavior (horizontal axis in Fig. 4), we compute the success rate by finding the frequency of trials that show the target behavior. We reset the robot randomly by sampling a starting pose in a validation set of trajectories. All error bars in simulation results (Figures 4, 5) are **1-sigma** error bars. The standard error is calculated by evaluating DynaGuide on six policies trained on individually separate train-validation splits of the Calvin dataset. Because each of the six success rates is already an average across 100 trials, we use the standard error of the mean as the sigma. We assume Normally-distributed errors, so the standard error is computed with the standard $\sigma/\sqrt{n}$ formula where $\sigma$ is calculated through a Numpy function. For the line plot (Fig. 5 Right), the shaded regions were computed as standard error of the mean of six success rates—the same as the bar graphs.

**ArticulatedParts Experiment**. To count as being sufficiently articulated, we require buttons and switches to be fully pressed/flipped such that the light changes state. For drawers and doors, they must be articulated past halfway from the starting location to their end location.

**MovableObjects Experiment**. In this task, we focused on the ability to steer the policy towards the colored cubes. To be successful, we required the robot to be touching the cube and displace it a slight distance, which can be possible by lifting the cube or nudging it.

**UnderspecifiedObjectives Experiment**. In this task, we used the **ArticulatedParts** experiments but fed all the guidance approaches with a lower quality guidance condition. Instead of taking the observations from the final states of validation trajectories, we manually set the target object to the desired pose (e.g. drawer open, or door to the left). We sample states for all the other objects, and we randomize the robot by setting it to a start pose sampled from the validation trajectories. The critical difference between these $\mathbf{g}^+$ and the $\mathbf{g}^+$ used in the other experiments is that the robot is no longer shown directly interacting with the object of interest. Even though the robot position is randomized, it is still possible to figure out the target behavior because it is the one object state that stays constant between conditions.

**MultipleObjectives Experiment.** This task, we extracted guidance conditions from validation trajectories that represented multiple types of behaviors. We are not interested in chaining the behaviors, but rather a steered policy that offer multiple desired behaviors with comparable frequency while avoiding the undesired behaviors. During evaluation, we still terminated the rollout after a single behavior and we compute the success rates in the same way as all previous experiments. However, instead of reporting a single success rate per task, we report the whole task distribution (Fig. 5, Left).

**UnderrepresentedBehaviors Experiment.** This task required retraining the base policies with modified training data. Using privileged state information to get trajectory behavior labels, we intentionally removed `Switch-On` behavior from the training set of the base diffusion policy to create sets where 1%, 2%, 5%, 10%, 20%, 40%, 60%, and 80% of the original `Switch-On` data was kept.

**BlockTouch Experiments**. We collect synthetic data of the agent navigating to a randomly selected square by creating a bezier curve with 0-2 intermediate points. This bezier curve means that the dynamics model can't infer the final destination with perfect accuracy based on the current direction of travel. Instead of using the metric $\mathbf{d}$, we directly train the dynamics model to classify the output square color as a 4-dimensional vector categorical distribution. During inference-time, we use a cross entropy metric between this distribution and a one-hot vector representing the desired square color. During training and all data collection, we randomize the location of the squares. During test-time, we introduce specific square arrangements to test the properties of the guidance In the *EarlyDecision* test, we still randomize the squares but ensure that the cubes stay in their own regions, forcing the agent to make an early directional decision. In *LateDecision*, we keep the squares close together, and in *Go to Furthest Cube*, we always have the blue square on the opposite side of the environment as the starting agent location. This agent must move past the three other squares to find the blue square. We terminate the environment once any square has been touched.

## B.5 Experimental Setup: Real Robot

**Base Policy**. We use a publicly available trained diffusion policy provided by the Universal Manipulation Interface repository (github link). It was trained on $\approx$2k trajectories of cup grasping, reorientation (to move the handle to the left), and placement on a saucer. The policy takes images from a gopro with a Max Lens mod and outputs 16 actions per generation. These relative actions represent the change in the robot from its current position [6]. This policy has mostly seen single cups and saucers in the environment, but when provided with multiple cups, it will choose a cup at random. To demonstrate DynaGuide on existing policies, we do not modify this policy. Although the policy offers random choice, it also has a bias towards the left side of the environment. To account for this bias, we randomize the location of the desired cup in **CupPreference**. We also place the computer mouse in the area opposite to this bias in **NovelBehavior** to ensure that all improvements are due to steering and not base policy bias.

**Data**. We train the dynamics model on the open-source cup rearrangement data provided by the Universal Manipulation Interface repository (data link). Because we want to steer the policy towards one of two cups, we need to train the dynamics model on these decisions. Using the Universal Manipulation Interface hardware, we collect 500 demonstrations that shows two cups and one saucer in the experiment scene. We pick a cup and place it on a saucer. We use different cups and saucers, with no correlation between cup pairs and saucers. We combine this data with the existing cup arrangement data for the dynamics model. For the **NovelBehavior** experiment, we additionally train the dynamics model on 3648 publicly available trajectories of computer mouse rearrangement [26], which gives it the experience needed to steer the base policy towards the computer mouse.

**Setup**. We use an ARX5 robot arm equipped with soft Fin Ray fingers and a gopro with Max Lens mod that mimics the setups used to collect base policy's training data. We use the hardware controller stack provided by the Universal Manipulation Intervace adaptation for ARX5 arms (Github Link) and a power supply with overcurrent protection for safety during deployment.

**Evaluations**. We conduct 20 trials for each base/guided policy. Real-world evaluations are very resource intensive, so we did not create confidence intervals. This is very common for real robot results, even for large projects [5, 57].

**CupPreference Experiment**. We arrange the red and grey cups such that the starting distance to either cup is roughly equidistant from the robot. We randomize this distance by sometimes placing the two cups close, and other times placing the two cups far away. We place the saucer randomly in the environment. For 50% of the trials, the red cup is on the left, and the other 50% the red cup is on right. We measure success as picking up a cup and placing it on the saucer.

**HiddenCup Experiment**. We arrange the red cup in front of the grey cup, such that the grey cup is always closer to the robot than the red cup. We randomize the distance between the red and grey cup, as well as the overall location of the two cups from the robot. We randomly place a saucer in the environment, although we count a cup grasp as a success without needing this saucer. We

discovered that the two cup setup in this configuration is out of distribution enough for the base policy to increase the mistake of cup placement. Because we are trying to steer for cup color preference, the act of grabbing the cup is sufficient for this experiment. The previous experiment tested full behavior success.

**NovelBehavior Experiment**. Like **CupPreference**, we arrange the red cup and black computer mouse equidistant from the robot's starting location. As previously mentioned, we place the mouse in the area opposite of its side bias to ensure that the observed effects are attributed to steering and not base policy bias. Like in **HiddenCup**, we count a grasp attempt of the computer mouse as a success. This is because the computer mouse is very out of distribution for this base policy and it will not successfully grab the mouse. What matters in this experiment is being able to reach for a novel object through dynamics guidance.

## C   Broader Impact

DynaGuide contributes to the field of robotics by improving the ability for pretrained policies to be steered without retraining, potentially reducing energy consumption otherwise needed during retraining. Adding steering onto pretrained policies also potentially increases accessibility for labs that would otherwise be unable to retrain large policies. This steering approach can also be an effective way of removing unwanted or problematic biases in trained robot policies after the training process. The dynamics model is trained on data that is either publically available or easy to collect, and the data contains no private or sensitive information.

Because DynaGuide allows steering of existing policies, there is a risk of bad actors steering off-the-shelf policies to do dangerous or malicious behaviors, a risk also present for other generative models. To mitigate these risks, trainers of the base policy can ensure that malicious behaviors are not represented at all. Knowing the mechanism of DynaGuide, it may also be possible to train base policies to act adversarially to the guidance signal if asked to do malicious behaviors, making the guidance as difficult as retraining the policy from scratch.

## D   Code, Assets, and Licenses

All our models are trained on publicly available data and leverages codebases and assets under these licenses:

- CALVIN Environment and Data (MIT License) [30]
- Robomimic Codebase (MIT License) [29]
- DinoV2 Model (Apache license) [37]
- Universal Manipulation Interface Codebase, pretrained model, and data (MIT License) [6, 26]

Our code will be publicly released under the MIT license in the near future. Provided in the supplemental material is a barebones version of the code.

