# OpenReview forum: "DynaGuide: Steering Diffusion Polices with Active Dynamic Guidance"
_NeurIPS.cc/2025/Conference — NeurIPS 2025 poster_

### Official Review · Reviewer_ma6B · 2025-06-05

**Clarity:** 3
**Significance:** 3
**Originality:** 3
**Rating:** 5
**Confidence:** 4

**Summary:**

The authors propose DynaGuide to steer pre-trained diffusion policies at inference time without needing to retrain them. It uses a separately trained dynamics model to predict the future outcome of an action. The goal is specified through positive and negative goal images. By calculating the distance in a latent space between the predicted outcome and the goal images, the system generates a guidance signal. The gradient of this signal is used to directly modify the action denoising process of the diffusion policy, steering it towards desired behaviors and away from undesired ones.

**Questions:**

- **Negative Goal Selection:** How sensitive is the method to the choice of negative goal images? For high-precision tasks, a failure state might look very similar to a success state, which could make providing effective negative guidance difficult.

- **Alternative Guidance Modalities:** As noted in the paper's future work, can guidance be extended beyond static images to other forms like language ("push the red block gently") or kinesthetic demonstration to provide more fine-grained control over the trajectory?

**Ethical Concerns:**

["NO or VERY MINOR ethics concerns only"]

**Final Justification:**

The author makes a great effort to clearify my concerns. I believe this paper would be a great fit for the conference. Thus, I maintain my score.

**Limitations:**

- **Lack of Trajectory Control:** The guidance mechanism steers towards a final state but offers no direct control over the path taken to reach that state.

- **Computational Cost:** To use strong guidance without destabilizing the trajectory, the method relies on MCMC sampling (running each denoising step multiple times), which increases the computational overhead at inference time.

- **No Online Learning:** The guidance is applied only at inference time and is forgotten afterward. The base policy does not adapt or improve from the guidance it receives.

**Quality:**

3

**Strengths And Weaknesses:**

### Strengths

- **Plug-and-Play:** It can steer existing, off-the-shelf diffusion policies without any modification or fine-tuning.

- **Robustness to Low-Quality Goals:** By separating the policy and the dynamics model, the system is more robust to "under-specified" goal.


### Weakness

- **Guidance is Outcome-Based:** The method can only specify the desired *final outcome* of a task via an image. It cannot control *how* the robot should perform the task (i.e., the specific trajectory or method of execution).

- **Reliance on Dynamics Model:** The performance is entirely dependent on the quality and data coverage of the dynamics model. If the dynamics model cannot accurately predict the consequence of an action, the guidance will be ineffective.

- **Limited Baselines:** The paper does not compare against methods that more tightly integrate a world model into the diffusion process itself, like the original diffuser paper + inpainting [1] , which could be a relevant alternative approach.

[1] Janner, Michael, Yilun Du, Joshua B. Tenenbaum, and Sergey Levine. “Planning with Diffusion for Flexible Behavior Synthesis.” arXiv, December 20, 2022. http://arxiv.org/abs/2205.09991.

---

> ### Author Rebuttal · Authors · 2025-07-28
>
> Thank you for your positive feedback and thoughtful analyses of DynaGuide. We appreciate your highlighting of Dynaguide’s robustness to lower quality goal conditions that would otherwise cause goal-conditioned baselines to fail. Below, we answer the questions and address the limitations.
>
> # Answers to Questions
> ## Sensitivity to negative goals
> Empirically, we didn’t have to cherry-pick the negative goals to get DynaGuide working on the multi-objective setup. Sampling the negative states as the final states in a randomly selected set of validation trajectories was sufficient.
>
> The performance of DynaGuide depends on the ability of the latent space to separate out desired and undesired outcomes. We use the DinoV2 latent space, as past work [1] had yielded successful results in the planning domain. If a particular task has similar success and failure states, then a special latent space should be constructed that separates them appropriately. This latent space will probably need to be trained in a task-specific way. The DinoV2 space was sufficient for our experiments, but improving the latent space is a meaningful direction for future work.
>
> ## Alternative guidance modalities
> As long as you can embed the guidance conditions into the latent space, any format of guidance condition should be valid. For example, if a CLIP embedding is used, then text can be used as guidance conditions as well as images. The latent space can also encode history and kinematic properties of the trajectory, if trained with the correct objectives. We leave the exploration of latent spaces to future work. The DynaGuide method is general and should work on these specialized latent spaces.
>
> # Discussion of Limitations and Weaknesses
> ## Guidance is Outcome-Based / Lack of Trajectory Control
> In the simplest formulation DynaGuide, we steer the policy with guidance conditions that represent outcomes. However, because Eq. 2 is a soft minimum, it is also possible to add intermediate guidance conditions that represent how a task should be done. We actually did this for the real robot setup due to the wrist camera’s end observation being not a good representation of the task. We found that these intermediate guidance conditions were critical in getting the desired performance improvement. DynaGuide as a framework supports guidance at any point(s) in the trajectory, as long as they can be represented as anchor points in a latent distribution.
>
> ## Reliance on Dynamics model
> We agree with this concern. By construction, DynaGuide relies on the dynamic model’s predictions for good guidance. However, the dynamics model can be trained on additional lower-quality and/or less-relevant data compared to an expert policy, making it easier to improve the robustness of the dynamics model compared to doing the same for the expert policy. In our Calvin experiments, we train the dynamics model on interactions in other domains (ABC split). In our real robot experiments, we add data from other object interactions. In both cases, the dynamics model benefited from this additional data.
>
> ## Limited Baselines
> There is indeed a line of work that integrates world models into the planning and diffusion process. Because we were focused on the ability to guide existing policies, we were less thorough in the planning-based baselines like DINO-WM and Diffuser. On the bigger picture, these approaches are still relevant and we will try to implement & test some of these approaches for the final publication.
> In the meantime we added the following baselines in response to other reviews
> + Adding DynaGuide-Sampling (GPC) to real world results (see: RYQt)
> + Reducing the dynamics model training dataset to `D` from `ABCD` (see: yJvQ and n1oB)
> + Increasing the data of the goal-conditioned policy to `ABCD` (see: yJvQ and n1oB)
> + Adding HER-like training to goal-conditioned policy (see: n1oB)
> + Testing classifier-free guidance on the multi-objective task (see: n1oB)
>
> ## Computational Cost
> The MCMC sampling is indeed adding to the computational overhead, although unlike sampling, DynaGuide’s computational cost is not dependent on the rarity of the target behavior or the difficulty of the task. When run on an RTX 3080 with a real robot, this inference process still runs in real-time due to the model predicting entire action chunks at once. For a profile of run times, see our response to Reviewer n1oB.
>
> ## No online learning
> It is true that the guidance is forgotten afterwards. However, the guidance is repeatable because the *guidance conditions are stored.* Guidance conditions can be added or removed dynamically based on past performance–and this is a notion of adaptation / persistence. This online learning may not be as expressive as finetuning, but it is more explainable, robust, and easily reversible if needed. It adds to the modular capabilities of DynaGuide.
>
> # Summary
> Thank you for your thoughtful observations and questions. We have done our best to provide answers to questions about DynaGuide’s limitations and possibilities for other modalities and applications. We welcome any follow-up questions or comments.
>
>
> [1] Gaoyue Zhou, Hengkai Pan, Yann LeCun, and Lerrel Pinto. DINO-WM: World models on pre-trained visual features enable zero-shot planning.

---

> > ### Comment · Reviewer_ma6B · 2025-08-03
> >
> > Thanks for the clarification. After the revision and new baselines, this paper would be a great fit to NeurIPS!

---

### Official Review · Reviewer_n1oB · 2025-06-22

**Clarity:** 3
**Significance:** 2
**Originality:** 2
**Rating:** 4
**Confidence:** 4

**Summary:**

This work introduces DynaGuide, a method for guiding diffusion policies towards desirable goals by using classifier guidance. The guidance mechanism includes a learned latent dynamics model operating on DinoV2 embeddings. The dynamics model predicts a future state from the current state and action chunk, and the gradient of the distance from the desired goal embedding is used for guidance. This method allows decoupling guidance from policy, allowing dynamics models to be swapped out as needed, and can be used with an off-the-shelf policy. Experiments on CALVIN tasks show that DynaGuide can help in cases where the goal is misspecified or corresponds to an underrepresented behavior of the base policy.

**Questions:**

- The guidance metric in Eq. 2 compares the latent future state $z_{t+H}$ with desired goal state $g$. This would only make sense if the goal $g$ is achievable within $H$ steps from the current state. Since this is generally not true, I am not sure why this provides a meaningful guidance signal. In fact, it may actually be detrimental since the agent can try to reach some unrealizable state to minimize distance to goal within $H$ steps.
- The value of $H$ is not explicitly mentioned, but appendix line 951 specifies that the policy produces action sequences of length 16 - does that mean that $H=16$ for all experiments? Why was this specific value chosen?
- Why can’t position guidance be used for the moving objects experiments? In [3], the authors use block stacking and kitchen rearrangement tasks - what makes the cube moving experiment fundamentally different from these tasks?
- The goal-conditioned baseline is trained by using the last observation of each trajectory as the goal. However, in lines 218-220, the authors mistakenly cite some papers which use goal relabeling techniques (such as hindsight experience replay [4]) to improve training of goal-conditioned policies. This has shown to work in visual domains as well [5,6]. Then, despite referencing such works, why were these techniques not used when training the goal-conditioned baseline?
- The dynamics model in Eq. 1 predicts a future state $H$ steps away from the current state and action chunk, which is tractable only for deterministic environments. I wonder if the authors have any thoughts on extending this approach to the stochastic setting?
- Typing error in Algorithm 1: the stochastic sampling loop does not correctly apply $M$ denoising steps at the same noise level $k$. See lines 6-9 in Algorithm 1 of [3].
- Typing error in Algorithm 1: line 1 should be “positive guidance conditions $g^+$, negative guidance conditions $g^-$”

*[3] Wang, Yanwei, Lirui Wang, Yilun Du, Balakumar Sundaralingam, Xuning Yang, Yu-Wei Chao, Claudia Perez-D'Arpino, Dieter Fox, and Julie Shah. "Inference-Time Policy Steering through Human Interactions." arXiv preprint arXiv:2411.16627 (2024).*

*[4] Andrychowicz, Marcin, Filip Wolski, Alex Ray, Jonas Schneider, Rachel Fong, Peter Welinder, Bob McGrew, Josh Tobin, OpenAI Pieter Abbeel, and Wojciech Zaremba. "Hindsight experience replay." Advances in neural information processing systems 30 (2017).*

*[5] Nair, Ashvin V., Vitchyr Pong, Murtaza Dalal, Shikhar Bahl, Steven Lin, and Sergey Levine. "Visual reinforcement learning with imagined goals." Advances in neural information processing systems 31 (2018).*

*[6] Florensa, Carlos, Jonas Degrave, Nicolas Heess, Jost Tobias Springenberg, and Martin Riedmiller. "Self-supervised learning of image embedding for continuous control." arXiv preprint arXiv:1901.00943 (2019).*

**Ethical Concerns:**

["NO or VERY MINOR ethics concerns only"]

**Final Justification:**

During the rebuttal, the authors added some meaningful goal-conditioned baselines missing from the paper and some preliminary results for classifier-free guidance, with the assurance that full results will be added in the updated paper. They also added wall clock comparisons and results using the same amount of data (CALVIN-D only) as the baselines. Some additional clarifications around the experimental setting and wording of certain claims were also provided.

The main concern that came up during the discussion was the method used for guidance, particularly whether it is sensible to compare $t+H$ observation with the end goal when it may not be attainable within the horizon. The authors clarified that this is indeed a limitation of the method, and in the general case (like the real robot experiment), the user may need to provide guidance at intermediate states. In my opinion, this limits the applicability of the method since this guidance is quite difficult to provide in practice.

Therefore, I maintain my rating leaning towards acceptance. I appreciate the fruitful discussion with the authors during the discussion period, which helped me better understand their method.

**Limitations:**

The authors have briefly discussed the limitations of their work with respect to the type of guidance provided and lack of any mechanism to guide how to achieve the desired outcome. There are several important limitations that have been left out, including the computational cost of DynaGuide compared to baselines, the fact that it seems limited to deterministic environments, and that the base policy must already be able to solve the task in order for guidance to elicit the desired behavior.

**Paper Formatting Concerns:**

The authors use much smaller fontsize for all figure/table captions and use negative spaces in several places (right after Figure 2 and Figure 6 being the most egregious examples). Both of these violate the NeurIPS formatting guidelines — from the call for papers, “Submissions that violate the NeurIPS style (e.g., by **decreasing margins or font sizes**) or page limits may be rejected without further review”.

**Quality:**

2

**Strengths And Weaknesses:**

**Strengths**

- The experiment design is done quite well. The authors study different settings designed to test different capabilities, such as working with low-quality goal conditions, multiple objectives, and testing underrepresented behaviors of the base policy.
- It is not common to see experiments on real robots, and the fact that DynaGuide works in this setting is a strong point in its favour.
- The authors have provided details of their implementation and baselines in the appendix, as well as provided their code in the supplementary material. While I did not study the code, the details in the appendix are quite thorough, and the authors’ efforts in promoting reproducibility are appreciated.
- The paper is written quite clearly and is easy to follow.

**Weaknesses**

- DynaGuide unifies classifier guidance with learned dynamics, which is new for robotics, but inherits much from prior dynamics-distillation work (DINO-WM) and inference-time steering (ITPS). The advance is incremental yet practical; clearer articulation of conceptual novelty over these works would strengthen the contribution.
- One primary motivation of DynaGuide is that typical goal-conditioned policies require anticipating the set of goals during training such that they cover the possible goals encountered during inference. DynaGuide does not seem to solve this problem, since the steering mechanism relies on the base policy already having the ability to solve the required task. Therefore, it seems that the claim in lines 34-42 that sets DynaGuide apart from existing methods should be reconsidered.
- In practical settings, it is rarely feasible to specify the exact goal observation that the agent needs to accomplish. In line 103, the authors claim that “Other methods with fewer assumptions…using future observations directly as the goal”. Assuming access to goal observations for guidance is a much stronger assumption than relying on goal specification via natural language, for example.
- The dynamics model uses much broader data than the policy (CALVIN-ABCD vs D). For truly off-the-shelf steering, users must still collect a large amount of unstructured interaction frames to fit a task-specific dynamics model.
- One essential baseline not included in this work is classifier-free guidance, which has been proposed in prior work for goal-conditioned policies [1,2] and works quite well. Like DynaGuide, this also allows specifying multiple positive and negative goals.
- DynaGuide’s stochastic sampling loop increases per-step computation by roughly $M\times$ relative to a conventional goal-conditioned policy, resulting in significantly slower inference. This overhead should be discussed in the paper. By contrast, the sampling baseline’s need for multiple forward passes per step is not as serious a drawback since those passes can run in parallel, so its wall-clock latency should actually be much less than DynaGuide’s inherently sequential guidance routine. Apart from a discussion, including some wall clock time comparisons during inference would strengthen the experiment sections.

*[1] Reuss, Moritz, Maximilian Li, Xiaogang Jia, and Rudolf Lioutikov. "Goal-Conditioned Imitation Learning using Score-based Diffusion Policies." In Robotics: Science and Systems. 2023.*

*[2] Ajay, Anurag, Yilun Du, Abhi Gupta, Joshua B. Tenenbaum, Tommi S. Jaakkola, and Pulkit Agrawal. "Is Conditional Generative Modeling all you need for Decision Making?." In The Eleventh International Conference on Learning Representations.*

---

> ### Author Rebuttal · Authors · 2025-07-29
>
> Thank you for the thorough comments, critiques, and questions. We appreciate your highlighting of the real robot experiments present in DynaGuide and the quality of experimental design. Below, we include additional experiments and discussion based on the points raised.
>
> # Addressing Main Limitation Remarks: Additional Experiments
> ## Lack of GC baselines like HER
> In the table below (additional results on exp. 3), we add `GC with HER`. To implement a HER-like algorithm, we train the GC policy on intermediate goals selected uniformly from a future observation in a trajectory. The results show some improvement on select tasks (notably the door task) but drops in other tasks. Similar results happen when the GC baseline is enriched with the `ABCD` data split (`GC with ABCD`). Overall, training on intermediate goals / enriched data had mixed results on performance. DynaGuide still consistently outperforms it. We will correct citations in the final publication.
>
> | Label         | button_on     | button_off    | switch_on     | switch_off    | drawer_open   | drawer_close  | door_left     | door_right    |
> |-|-|-|-|-|-|-|-|-|
> | DynaGuide [in paper]    | 0.33±0.031    | 0.36±0.032    | 0.24±0.017    | 0.26±0.024    | 0.26±0.015    | 0.27±0.017    | 0.25±0.018    | 0.19±0.015    |
> | Original GC [in paper]  | 0.023±0.0080  | 0.053±0.015   | 0.073±0.018   | 0.047±0.0067  | 0.00±0.00     | 0.0033±0.0033 | 0.12±0.012    | 0.080±0.016   |
> | GC with ABCD [new]  | 0.054±0.017   | 0.10±0.012    | 0.064±0.015   | 0.047±0.015   | 0.00±0.00     | 0.00±0.00     | 0.095±0.026   | 0.075±0.021   |
> | GC with HER [new]   | 0.0067±0.0042 | 0.087±0.030   | 0.027±0.011   | 0.020±0.0073  | 0.00±0.00     | 0.00±0.00     | 0.21±0.023    | 0.13±0.021    |
>
> ## Wall Clock comparisons of DynaGuide
> We have included some wall time analyses below on a 3090 GPU with no additional optimizations.
> + DynaGuide. Setup: 4 stochastic sampling steps, 10 inference denoising steps (DDIM). Time for inference (for chunk of 16 actions): **1.2±0.08 seconds**
> + DynaGuide-Sampling. Setup: 5 samples, batched. Time for inference (for chunk of 16 actions): **1.05±0.02** seconds
> + Base policy / Goal Conditioned. Time for inference (for chunk of 16 actions): **0.09±0.01 seconds**
>
> When batched, DynaGuide-Sampling runs marginally faster than DynaGuide due to fewer runs through the dynamics model. However, for rare behaviors, the sampling baseline must run many samples to get a desired behavior. Scaling this through batching would require larger memory. In contrast, DynaGuide does not require any additional computation or memory scaling to search for underrepresented behaviors successfully (Exp. 5).
>
> ## Dynamics Model Gets More Data
> This is a concern shared by reviewer RYQt. In response, we have run DynaGuide using just the `D` split. Reducing the dynamics model dataset does reduce performance, as expected. However, as seen in the GC table above, adding the ABCD split to the goal-conditioned baseline does not improve performance much. Therefore, the dynamics model makes better use of the additional data compared to the GC baseline.
>
> | Label         | button_on     | button_off    | switch_on     | switch_off    | drawer_open   | drawer_close  | door_left     | door_right    |
> |-|-|-|-|-|-|-|-|-|
> | ABCD    [from paper]      | 0.70±0.031    | 0.70±0.038    | 0.74±0.027    | 0.76±0.037    | 0.60±0.015    | 0.66±0.029    | 0.74±0.025    | 0.68±0.026    |
> | D SPLIT ONLY  [new] | 0.66±0.023    | 0.69±0.038    | 0.54±0.043    | 0.58±0.031    | 0.51±0.022    | 0.53±0.024    | 0.65±0.019    | 0.57±0.036    |
>
> ## Classifier-free Guidance
> This is a reasonable baseline for the multi-objective setup. We implemented CFG and provide some basic results below for multi-objective, focusing on two tasks in Exp. 4. We tuned CFG’s guidance strength using a grid search to obtain these values.
>
> | Button On or Switch On    | Button | Switch |
> |-|-|-|
> | DynaGuide [paper]  | 0.54±0.02   | 0.26±0.04   |
> | CFG [new]  | 0.18±0.015   | 0.37±0.017   |
>
>
> | Door Left or Button On   | Door | Button |
> |-|-|-|
> | DynaGuide [new]  | 0.19±0.023   | 0.54±0.03  |
> | CFG [new]  | 0.28±0.032   | 0.37±0.0084  |
>
> As you suggested, CFG can follow multiple objectives. However, its average per-task success (0.3) is lower than DynaGuide (0.38). One possible reason for this is the averaging of signals from the conditional models in the CFG implementation. In contrast, DynaGuide’s distance-based log-sum-exp prioritizes objectives that are the most reachable, leading to better performance with conflicting objectives.
>
> # Other Main Remarks and Questions
> ## Novelty of Contribution
> DynaGuide uses classifier guidance like ITPS, but the guidance signal is propagated through a dynamics model instead of euclidian distance in the action space. DynaGuide uses a latent space like DINO-WM, but instead of running model-based planning, it modifies the behaviors of a base policy. DynaGuide also consolidates the latent distances of multiple objectives (eq. 2), which is not possible in DINO-WM.
>
> In general, DynaGuide builds upon DINO-WM and ITPS, but combining these approaches required distinct innovations. The final result yielded a novel approach to guiding diffusion robot policies. We also provide real robot results, which were not provided by DINO-WM.
>
> ## Improper comparison of future state to desired end state
> This is a valid concern about the method. In the CALVIN environment, early task disambiguation allowed us to let $t+H$ be the last state of the trajectory (mentioned in Lines 149-150; this will be clarified in the final publication). Because the dynamics model predicted the final state, it was directly comparable to the desired final states. This trick was also used in DINO-WM [1].
>
> On the real robot experiments, the longer horizon and the wrist-mounted camera meant that $t+H$ could no longer be the final state. To account for the exact issue raised by your comment, we sampled states throughout desired behavior trajectories as the guidance conditions. The log-sum-exp acts like a soft minimum for the latent distances, so it automatically downweights states that are too far away / unreachable.
>
> These details are somewhat hidden in the current paper version so we will add more details in the final publication.
>
> ## Expanding DynaGuide to stochastic environments
> The current dynamics model is best for deterministic environments, but Eq. 1 can also be interpreted as a stochastic log-probability loss with unit variance. To properly adapt the dynamics model to a stochastic environment, we would need to make the variance learnable, i.e. getting the model to provide a measurement of uncertainty through specialized architecture like a VAE. This is an interesting direction for future work.
>
>
> # Other Remarks
> ## Lack of position guidance baseline in moving object experiment
> Position Guidance requires knowing the exact 3d location of an object to define a euclidian distance objective for use during steering. In ITPS, a human provided this 3d location in every environment rollout, which enabled its use on movable objects. In all our baselines, we don’t assume privileged state knowledge. For fixed objects, we get their locations by using the final gripper positions in the training data. This heuristic is not available for movable objects.
>
> ## Choice of action horizon value
> This action prediction horizon value is 16, and while we never conducted extensive experiments for this value, we found that the guidance is weaker for shorter values like 8 and the diffusion policy performs worse for very long values like 24 or 32, so we picked 16 as a balancing point.
>
> ## DynaGuide requires base policy to have ability to solve target task
> For steering approaches, it’s generally assumed that the base policy has the skills to achieve a behavior. Otherwise, the base policy will not be of much use. The referenced claim in lines 34-42 focuses on assuming knowledge of the *steering objective*, which is necessary for goal-conditioning because it is trained on the steering signal. In contrast, DynaGuide can handle *novel steering objectives* that leverage existing skills.
>
> An elucidating example: without using CFG, our goal conditioning won’t be able to *avoid* a behavior unless we train it specifically. However, we *can* avoid a behavior with DynaGuide by negating the guidance signal. This is an example of an *unanticipated steering objective* being applied to *known skills*. This should be clarified in the paper and we will do this in the final publication.
>
> ## Goal Assumptions
> >  Assuming access to goal observations for guidance is a much stronger assumption than relying on goal specification via natural language, for example.
>
> Yes, this is absolutely true. Although, the data needed to train a language-conditioned model requires more assumptions (task segmentation, good task labels, etc). We were having this interpretation in mind for line 103, although we see the ambiguity and will correct accordingly.
>
> ## Typing Errors in Algorithm 1
> Thanks for finding these typing errors. We will fix them in the final paper.
>
> ## Formatting concern
> Based on our observation of past NeurIPS papers, our interpretation of the CFP remarks was that they referred to main paper margins / main text size. We are happy to change the caption formatting if it was the incorrect interpretation.
>
> # Summary
> Thank you for the very detailed review and thoughtful questions. We have done our best to answer them by providing additional results including wall clock analysis, results with a different data split, stronger goal-conditioned baselines, and others. We also answer other conceptual questions and provide some discussion. We can provide further clarification if anything was not clear.
>
> [1] Gaoyue Zhou, Hengkai Pan, Yann LeCun, and Lerrel Pinto. DINO-WM: World models on pre-trained visual features enable zero-shot planning.

---

> ### Comment · Reviewer_n1oB · 2025-08-05
>
> Thank you for the response. The GC baselines using HER and the experiments with CALVIN-D are appreciated.
>
> > Wall Clock comparisons of DynaGuide
>
> The appendix specifies that the denoiser used 100 steps for all experiments (line 984). Therefore, the wall clock times in the rebuttal do not consider the same setting as the experiments in the paper. I request the authors to perform this analysis using the same parameters as the experiments to ensure fairness.
>
> > Classifier-free Guidance
>
> My comment meant that CFG is a natural baseline for all settings, and can also handle multi-objective guidance like DynaGuide. In general, CFG is the preferred method for conditional sampling using diffusion models over classifier-based guidance (like DynaGuide), so I believe this is an essential baseline, which should have been part of the paper.
>
> > Improper comparison of future state to desired end state
>
> Thank you for the clarification. If I understand correctly, if some tasks do not show early task disambiguation, then a similar procedure of sampling intermediate states from desired behavior trajectories must be used? This will be a much stronger signal for guidance (since now guidance is provided every H steps rather than only at the goal state). This raises questions about general applicability of DynaGuide beyond tasks like CALVIN, and how it compares against other methods if it uses the same guidance signal as other methods (only the goal state).
>
> > Lack of position guidance baseline in moving object experiment
>
> Perhaps I am missing something, but why can't the end gripper position from the dataset be used to get the position of movable objects?

---

> ### Author Response · Authors · 2025-08-05
> **Continuing discussion**
>
> Thank you for your comments and additional points.
> > Wall Clock comparisons
>
> The lines referenced in the appendix (B1-Dynamics model) references the scheduler used to train the Dynamics model, not the denoiser of the diffusion policy. As is standard for diffusion policies, we train on 100 denoising steps but run inference on 10 denoising steps through the DDIM scheduler. All the results obtained in the paper have been done according to this DDIM 10-step scheduler, so we maintain that the wall clock results are correct and fair.
>
> > Classifier-free Guidance
>
> We agree that CFG is a meaningful general baseline and we will add a full sweep of these numbers in the final publication. We also agree that CFG is more standard than classifier-based guidance in generative modeling outside of robotics. However, in robot-learning, an external guidance signal provides distinct advantages, including additional expertise and an explicit understanding of dynamics. In contrast, with CFG this understanding is implicit. The vanilla CFG implementation also lacks the ability to explicitly shape guidance signals based on their reachability, which is enabled with the latent distances of DynaGuide. These challenges are less present in non-robot settings, and quality data is also more abundant for training CFG as compared to the robot setting. These factors might be the cause behind the lower performance of the CFG baseline in the robot setup, although we will have stronger conclusions once we get more CFG results.
>
> > Improper comparison of future state to desired end state
>
> Thanks for continuing this discussion. Yes, for the real robot experiment we used a uniform sampling of intermediate states in the trajectory as guidance conditions due to the wrist camera giving partial observability.   In general, DynaGuide works for both endpoint guidance and intermediate point guidance due to the log-sum-exp of the latent distances that provide a soft reachability selection for guidance signals. Therefore, DynaGuide is not only limited to tasks with early disambiguation like CALVIN. As an immediate example, the toy task (A.1) addresses these questions about early and late disambiguation. There, the base policy is trained on spline curve data with up to 3 turn points, meaning that the guidance must happen continually to reach the correct color. The `Hidden Cup` experiment in the real robot also shows a late decision example. DynaGuide is still able to meaningfully guide the agent towards a desired color (Figure 7) and cup (Figure 6). In our CALVIN experiments, we intentionally only use the goal state to keep it fair with goal conditioning and other methods that only used end state conditioning.
>
> > Lack of position guidance baseline in moving object experiment
>
> For stationary objects, the end gripper position in the dataset will be on average the location of that object. However, the movable objects are randomly shuffled between environment instances (i.e. between resets). The end gripper position on those tasks will always be on those movable objects, but *the locations will be randomized*. Therefore, the average gripper position is no longer meaningful. I think the movable objects may have been a bit of a misnomer / confusing term to use on our part. What we explicitly mean is this: *stationary objects*--things like cabinets, buttons. Does not move between environment resets. *movable objects*--things like the colored blocks. Placed randomly between environment resets. We investigate these objects as a means of showing conceptual guidance as opposed to place memorization. These results are supported with our real robot results and toy agent experiment.
>
> Thanks again for the response, and we welcome additional discussion!

---

> > ### Comment · Reviewer_n1oB · 2025-08-05
> >
> > Thanks for the additional clarification.
> >
> > Please mention in the implementation details that denoising is performed with 10 steps. I agree with the points about CFG, and that explicit guidance has certain benefits. The authors have indicated their intent to add it as a baseline in all settings.
> > Thanks for explaining the movable object setting, please add this in the paper/appendix to make the difference between the settings more clear.
> >
> > Regarding intermediate vs endpoint guidance: if I understand correctly, the experiments in A.1 are using intermediate state guidance? If so, the authors should explain it clearly since currently, it is not clear from the text that DynaGuide uses intermediate state guidance in this task. And my comment was not implying that DynaGuide is only limited to tasks with early disambiguation - rather it was about DynaGuide requiring much stronger guidance signal to solve general tasks. A series of intermediate states can be very hard to provide in practical cases, which is why goal-conditioned RL is generally framed as only providing the final goal state (nothing is stopping the user from specifying a series of intermediate sub-goals, but that is not desirable, hence the standard setting only provides the final state for guidance).
> >
> > To summarize into a concrete question - how will DynaGuide compare with standard methods using only the goal state for guidance for general tasks, where there may not be early task disambiguation?

---

> > > ### Author Response · Authors · 2025-08-05
> > > **continuing discussion**
> > >
> > > We will add the 10 step detail in the implementation--thanks for bringing it to our attention! We will also clarify the movable object setting terminology in the appendix for the final publication.
> > >
> > > We agree that intermediate states can be hard to provide in practical cases. On the real robot, we collected them from a few task demonstrations, which we acknowledge may not always be present. We clarify here that A.1 is *actually not using intermediate state guidance*. We used A.1 and `Hidden Cup` as an example in the above discussion to showcase DynaGuide's ability to deal with late disambiguation using end state guidance (A.1) and intermediate state guidance (`Hidden Cup`). Having to make the choice between end-state & future state is indeed a limitation of DynaGuide with respect to goal-conditioned RL approaches. Unlike goal-conditioned RL approaches which are trained with the steering objective in mind, DynaGuide is supposed to work on off-the-shelf policies. The tradeoff perhaps is the need for more careful guidance.
> > >
> > > This being said, there is nothing theoretically preventing the dynamics model from always being trained to predict the last state. However, for most dynamic model training formulations, the future prediction isn't propagated like Bellman backups, which might lead to degradation in performance for very long horizon dependencies. This is a limitation of the dynamics model and not of the core DynaGuide method. Future works might resolve this by training a dynamics model to respect future uncertainties in the prediction of end states. These dynamics models can be used under the DynaGuide formulation without much modification.
> > >
> > > In our case for the real robot experiments, our motivation for intermediate guidance conditions was not due to early / late disambiguation (the `Cup Preference` task is early disambiguation) but rather the quality of the eye-in-hand camera. The image state of the final state always had either a red or a black cup filling up the whole frame. Combined with a relatively clean training data, it means that the dynamics model could shortcut a lot of its inference and did not learn properly. If instead the robot were trained on third-person camera or used a state representation that included more information, we hypothesize that we wouldn't have needed the intermediate guidance.
> > >
> > > The toy results in A.1. is a starting point to answering your explicit question, which does pit DynaGuide against a late disambiguation task with only end state conditioning. But we acknowledge that there might be a more interesting additional experiment set that fully answers this question both qualitatively and quantitatively.
> > >
> > > Thanks again for your continued feedback and discussion. It is helping us improve the clarity of our paper and continuing to refine the capabilities and limitations of DynaGuide.

---

> > > > ### Comment · Reviewer_n1oB · 2025-08-06
> > > >
> > > > Thank you for the detailed response. This answers my question about intermediate vs endpoint guidance.
> > > >
> > > > While I agree that in principle, if a dynamics model can reliably predict the end state, then DynaGuide can use these predictions to guide the policy towards desired goals. Since this is not the case in general, it makes the method somewhat less principled, since my original question about why $z_{t+H}$ predictions are used to compute distances with a goal state that may not be attainable within $H$ steps still stands.
> > > >
> > > > Considering the above issue, I maintain my current rating leaning towards acceptance. I appreciate the prompt responses and clarification given by the authors.

---

> > > > > ### Author Response · Authors · 2025-08-08
> > > > > **Thank you for the discussion**
> > > > >
> > > > > Hi, and thanks again for this discussion. We respect your decision to maintain your current rating, and we will make one concluding comment / summary that hopefully encapsulates the intermediate vs. endpoint guidance discussion.
> > > > >
> > > > > The original concern raised was about comparing the dynamics model outputs to goal representations in the guidance conditions. *If the dynamics model only predicts $t+H$ steps in the future, what if the goal is not attainable in this number of steps?* This is a valid concern, and here is how we're currently addressing it in our paper.
> > > > >
> > > > > For CALVIN, we directly train the model to predict the final state, i.e. $t+H = T$ where $T$ is the horizon. The dynamics model is stable on this setup in CALVIN and the comparisons to goal-conditioning and other baselines are fair. However, this raises the question: what if we can't disambiguate the final state at earlier states? Or, if the end state is not an informative representation of the trajectory?
> > > > >
> > > > > We start to experiment with this exact challenge in the real robot experiment, where we let $H = 48$ steps into the future. To make this work, we needed to add intermediate points for guidance which were sampled from 5 trajectories that showed the correct task. This showcases the ability for DynaGuide to utilize intermediate guidance. As noted in the review comments, we acknowledge that not all tasks may have collected trajectories like this. This is a limitation of DynaGuide that shows up in tasks that have high ambiguity and therefore require dynamics models to be trained to predict intermediate trajectory states.
> > > > >
> > > > > In these cases, it may be possible to rely on the structure of the latent space to have an intermediate-state dynamics model and use only end-state guidance conditions. Even if there is a mismatch where $t + H <T$, the latent distance between a future state and a desired end state may still provide a meaningful signal. A good experiment to conduct would be to train the dynamics model in CALVIN or another environment on a fixed $H$ and run guidance using desired end states only. We hypothesize that the performance would be worse but still better than baseline. We will add these results in the appendix of our final publication.
> > > > >
> > > > > We will make the changes you suggested in your review and the above discussion. Once again, thanks for making this a fruitful discussion.

---

> > > > > > ### Comment · Reviewer_n1oB · 2025-08-08
> > > > > >
> > > > > > Thanks for this summary - it nicely wraps up our discussion.
> > > > > >
> > > > > > The current rubric for rating papers makes it quite hard to give a score of 5, as it seems that papers need to be essentially flawless with high impact, according to the description. My "ideal" score would be something like 4.5 based on the current rating system.

---

### Official Review · Reviewer_yJvQ · 2025-06-24

**Clarity:** 4
**Significance:** 4
**Originality:** 3
**Rating:** 5
**Confidence:** 4

**Summary:**

DynaGuide proposes to guide diffusion based robot policies by learning an independent dynamics model which influences the gradient field of the diffusion model.

This approach is evaluated on three simulated tasks using CALVIN and a real-robot task and compared to goal-conditioning and position guidance approaches.

In addition to help guide achieving desired behavior, this approach works with pretrained, off-the-shelf policies, with many potential applications for diffusion based robot learning.

**Questions:**

## Main Question:
Regarding CALVIN Datasplit (see Weaknesses). I do not believe running the full ABC-D ABCD-D CALVIN benchmark is reasonable to demand as a reviewer. I would welcome any results with the DynaGuide model trained only on D, as the other policies, if results are available.

Overall I am already very impressed with the method and hope that answering these questions will lead to even higher impact in the research community.

### Minor Questions to help improve my understanding:
- I am confused by the CALVIN underspecified objective. I assume any highlighting in Figure 3 is just for illustrative purposes? So what does it mean to "randomize robot states (and other states)"? Because I am quite surprised by goal-conditioning's performance drop.
    - Could DynaGuide guidance counteract BC's performance drop? Such an experiment would also more nicely highlight DynaGuide capabilities to work with different, "off-the-shelf" policies.
- On multi-objective guidance: Am I understanding correctly, that each objective is used to produce a guidance signal independently, which is then combined across all objectives, for example by addition? Or does the Dynaguide model actually receive a set of objectives at once?

**Ethical Concerns:**

["NO or VERY MINOR ethics concerns only"]

**Final Justification:**

My questions have been addressed and additional CALVIN D-D provided.
I recommend to accept this paper.

I believe investigating the transferability of the dynamics model by performing ABC-D experiments could further strengthen the potential impact.

**Limitations:**

yes

**Paper Formatting Concerns:**

None.

**Quality:**

4

**Strengths And Weaknesses:**

### Strengths
- The paper is very well written and easy to understand.
- Solid motivation and theoretical foundation.
- Multiple experiments in simulation and on real robot.

### Weaknesses
**I am worried about the data split for the experiments**. The details are a bit hidden in Appendix B4:
- The base policy is trained only on the CALVIN-D dataset, but the dynamics model is trained on ABCD. As the other baselines are not detailed, I have to assume they were also only trained on CALVIN-D.

    I am not entirely convinced that this is a good basis for a fair comparison regarding the generalization capabilities of the different baselines, as the guiding model has seen more data. I would welcome some discussion on this issue.

- Additionally, I would welcome an evaluation in the more common settings of ABC-D and ABCD-D (or D-D) on CALVIN.
This could help to estimate if the guiding model benefits from being trained once on a large dataset and guiding specialized policies, or alternatively, if both models basically require access to the same dataset anyway, to also be trained on the same data.

---

> ### Author Rebuttal · Authors · 2025-07-28
>
> Thank you for your positive response, detailed feedback about the CALVIN environment, and clarification questions about our experiments. We appreciate your recognition of DynaGuide’s versatility and our results in both simulation and real setups. Below we address your questions and add additional experiments inspired by your comments.
>
> # Dataset Splits: ABCD vs. D on Calvin (additional experiment)
> Yes, you are correct that the base model is trained on `D` and the dynamics model is trained on `ABCD`, which contains more data. We used the ABCD split to showcase the performance of DynaGuide with a dynamics model trained as well as possible. This is how we ensured fair comparisons in our paper results:
> + Comparing DynaGuide to DynaGuide-Sampling, we used the exact same base policy and dynamics model, varying only the steering method (sampling vs. guidance)
> + Comparing DynaGuide to Position Guidance, we used the exact same base policy and steering method, varying only the steering signal (dynamics model vs position euclidian distance).
> + Comparing DynaGuide to Goal Conditioning, we trained the base policy and goal-conditioned model on the same dataset and used the same guidance conditions.
> + Comparing DynaGuide to base policy, we used the same base policy *inside* of DynaGuide, meaning that performance improvements are explained solely by the guidance.
>
> We agree that training the dynamics model on the `D` split is a meaningful additional experiment because it completely isolates performance improvements of DynaGuide from data quality.  Below are the results on Exp. 1 with ABCD (paper results) and D splits for the dynamics model (requested)
> | Label         | button_on     | button_off    | switch_on     | switch_off    | drawer_open   | drawer_close  | door_left     | door_right    |
> |---------------|---------------|---------------|---------------|---------------|---------------|---------------|---------------|---------------|
> | ABCD  [in paper]        | 0.70±0.031    | 0.70±0.038    | 0.74±0.027    | 0.76±0.037    | 0.60±0.015    | 0.66±0.029    | 0.74±0.025    | 0.68±0.026    |
> | D [new] | 0.66±0.023    | 0.69±0.038    | 0.54±0.043    | 0.58±0.031    | 0.51±0.022    | 0.53±0.024    | 0.65±0.019    | 0.57±0.036    |
>
> As expected, the dynamics model performs worse when trained on a smaller dataset. However, in practical implementation, the dynamics model may be trained on a more diverse dataset than the base policy because it requires fewer quality assumptions about data than the base policy. Therefore, we can ask another question: does DynaGuide make better use of a larger data split than a goal-conditioned baseline? To answer this question, we train the goal-conditioned baseline on the `ABCD` instead of just the `D` split shown in the paper. Below are the results on Exp. 3 (which originally shows a large impact in goal-conditioned performance.)
>
> | Label         | button_on     | button_off    | switch_on     | switch_off    | drawer_open   | drawer_close  | door_left     | door_right    |
> |---------------|---------------|---------------|----------------|----------------|----------------|----------------|----------------|----------------|
> | DynaGuide [in paper]     | 0.33±0.031    | 0.36±0.032    | 0.24±0.017    | 0.26±0.024    | 0.26±0.015    | 0.27±0.017    | 0.25±0.018    | 0.19±0.015    |
> | GC with D [in paper]   | 0.023±0.0080  | 0.053±0.015   | 0.073±0.018   | 0.047±0.0067  | 0.00±0.00     | 0.0033±0.0033 | 0.12±0.012    | 0.080±0.016   |
> | GC with ABCD [new]  | 0.054±0.017   | 0.10±0.012    | 0.064±0.015   | 0.047±0.015   | 0.00±0.00     | 0.00±0.00     | 0.095±0.026   | 0.075±0.021   |
>
> Notice how the goal-conditioned model benefits slightly from additional data but still is far away from DynaGuide’s performance on Exp. 3. This showcases that DynaGuide can make better use of this additional data split compared to the goal-conditioned baseline.
>
> Conclusion from these two additional experiments: although the dynamics model degrades performance when trained only on D (DynaGuide w/ D exp), it leverages this additional data better than the goal-conditioned baseline (GC w/ ABCD exp).
>
>
> # Calvin Underspecified Objective Question
> > I am confused by the CALVIN underspecified objective. I assume any highlighting in Figure 3 is just for illustrative purposes? So what does it mean to "randomize robot states (and other states)"? Because I am quite surprised by goal-conditioning's performance drop.
>
> Yes you are correct that the highlighting / greyscaling in Fig. 3 is purely for illustration. In reality, in Exp. 3 we supply the methods with guidance conditions (goal images) that are out of distribution by showing the desired *environment state* but not the correct *robot state*.
>
> Concretely, for all other experiments, the guidance condition shows the robot interacting with the target object because the guidance condition image is taken after the robot accomplishes the task. For exp. 3, we still show the target object in the desired configuration but we show the robot in a random position (randomly selected state in a validation dataset). As an example: normally for the `drawer-open` task, the guidance condition shows the robot holding onto the open drawer by the handle. In Exp. 3, the guidance condition shows the same open drawer, but the robot is not grabbing the handle.
>
> Because all the models (dynamics, goal-conditioned) were trained with the robot on the target object, these guidance conditions in exp. 3 are generally out of distribution. This is why the goal-conditioned model's performance drops significantly. DynaGuide does not drop as much because these guidance conditions are still in-distribution to all individual components of the method; the guidance signal just becomes noisier. The guidance signals can still pull the policy in the direction of future states with open drawers, for example.
>
> We conducted Exp. 3 to showcase how the DynaGuide process can handle these out-of-distribution guidance conditions better than a model trained end-to-end on the goal distribution. Especially when we are trying to steer an off-the-shelf policy in a new environment with our own guidance conditions, this robustness is especially important. We are happy to clarify further if this response does not fully answer your confusion.
>
> # Counteracting BC Performance drop
>
> > Could DynaGuide guidance counteract BC's performance drop? Such an experiment would also more nicely highlight DynaGuide capabilities to work with different, "off-the-shelf" policies.
>
> Assuming that the dynamics model is still in distribution, it would be reasonable to hypothesize that DynaGuide can improve BC performance when the base policy is out of distribution. We base this hypothesis on the results of Exp. 5, where DynaGuide can pull a very underrepresented behavior from a base policy. Executing this behavior is out of distribution for the base policy due to intentionally removing that part of its training distribution. DynaGuide is still able to coax the base policy to execute that behavior.
>
> We also see an example of this on the real robot setup, where DynaGuide is able to get the robot to deliberately search for a red-colored cup when the base policy defaults to going towards the closest cup (Exp 6, Fig. 6 middle plot).
>
> In general, if the dynamics model is robust, then DynaGuide could be a tool for counteracting BC performance drop. To say this with more certainty, however, more experiments are needed.
>
> # Multi-objective guidance
>
> > On multi-objective guidance: Am I understanding correctly, that each objective is used to produce a guidance signal independently, which is then combined across all objectives, for example by addition? Or does the Dynaguide model actually receive a set of objectives at once?
>
> So DynaGuide actually takes a collection of guidance conditions and aggregates the distances into a single metric (Eq. 2). This seems like the latter approach in your question. This means that we only need to backpropagate through the dynamics model once per diffusion step. It technically is possible to compute guidance signals (action gradients) independently and aggregate them after backpropagation, although it is less computationally efficient.
>
> # Summary
> Thanks for your comments and feedback. We add additional results using a dynamics model trained on the `D` CALVIN split as well as a goal-conditioned model trained on the `ABCD` CALVIN split to further isolate the impacts of individual DynaGuide components. We also address various questions about the method. If any of your concerns were not addressed by this response, we welcome further discussion.

---

> > ### Comment · Reviewer_yJvQ · 2025-08-02
> >
> > Thank you for the detailed response.
> > The additional experiments and explanations are much appreciated and help me better understand the method.

---

### Official Review · Reviewer_RYQt · 2025-07-03

**Clarity:** 3
**Significance:** 3
**Originality:** 3
**Rating:** 4
**Confidence:** 3

**Summary:**

The authors present DynaGuide: a method for steering a base diffusion policy to desired task outcomes using the gradient feedback from an additional predictive dynamics model which matches positive and negative goal visual references.

The overall approach is illustrated in Figure 2. An additional dynamics guidance model is included to predict future latent state outcomes (trained by equation 1, page 4). Using these predicted outcomes, a denoising guidance signal is derived (equation 3, page 5) based on the action gradient with respect to a guidance metric $\mathbf{d}$ (equation 2, page 5). The guidance metric encodes positive $\mathbf{g^+}$ and negative $\mathbf{g^-}$ image references for visual scenarios to attain and avoid respectively. The authors hypothesize that their novel steering approach is well suited to finding under-represented samples and meeting multi-objective scenarios.

__Novelty and Contributions:__
- Novel guidance signal approach for diffusion models using gradient from a dynamics prediction model to steer the denoising process
- Versus existing baselines, their method boasts improved performance in achieving multi-goal objectives and under-represented behaviours.

__Experiments:__

A number of experiments explore the performance and properties of DynaGuide using a robot arm in the CALVIN environment [30].

In experiment 1 (section 4.1, figure 4 top) their approach fails to beat a more conventional goal-conditioned policy when interacted with standard articulated parts, however in experiments 2 (section 4.1, figure 4 bottom left) and 3 (section 4.2, figure 4 bottom right), their method significantly outperforms other baselines for movable and under-specified objects. The authors assert that this is because their method is more robust to situations that would normally be out-of-distribution for a normal goal conditioned policy given that goal, observation, and action inputs are all individually in-distribution to their modular setup while also using additional guidance signals in the latent space (described on lines 278-285).

In experiments 4 (section 4.3) and 5 (section 4.4) they also illustrate how their method is suitable for multi-objectives and its ability to find under-represented scenarios versus a sample based DynaGuide variant (GPC) as the sample-based method is restricted by the frequency of the desired behavior in the dataset being rare in this case.

Finally, in experiment 6, they test their method with a real robot versus the base policy. While the base policy is limited to the representative behaviors in the dataset, their policy effectively steers towards the conditioned goal.

**Questions:**

- Can the authors clarify why additional baselines were not included in the real-world experiment section (experiment 6, section 4.5)?

**Ethical Concerns:**

["NO or VERY MINOR ethics concerns only"]

**Final Justification:**

After reading reviewing the authors' rebuttal - and absent any further modifications - I have decided to keep my overall score the same (`Borderline Accept`). Please see my response to the authors' review for more details.

In short, my primary criticism is that I would of liked to see more experimental baselines applied to the real-world experimentation section. More experimental rigor seems to have been applied to the simulation section which ideally could be carried over to real-world experiments. Nonetheless, this is a solid paper but I feel requires some additional experimental testing to push it to the `Accept` level.

The authors have also included more analysis into hyperparameter selection which eases some of my concerns for this previous weakness item.

Please see my original review for a summary of the strengths and weaknesses of the work.

**Limitations:**

Yes. Limitations are listed in section 5.1 (lines 353-360).

**Quality:**

3

**Strengths And Weaknesses:**

__Strengths:__

- __Novelty and performance (originality and significance):__ Relatively novel and modern approach adapting recent diffusion models to a planning context using steering with primary advantages in executing multi-goal objectives and under-represented behaviours. Method outperforms state of the art baselines. See __Novelty and Contributions__ section above for more details.
- __Pertinent use cases (significance):__ As discussed in the experimental section, the proposed method excels in multi-goal and under-represented behaviours. The usage for under-represented behaviour may be especially important for real world applications such as robotics where training data may be more difficult to gather and thus some behaviours may have limited representation in the dataset.
- __Experimental rigour (quality):__ Experiments are done well for the most part (quality): Various aspects of the model were tested and modern relevant baselines compared against. Error bars are included. Through various experiments, they well illustrate the advantages of the proposed method compared to existing approaches for multi-goal objectives and under-represented behaviours (see __Experiments__ section above for more details). Ablation and parameter sweep results can be found in appendix A.2 with intuition concerning the parameter selection. However, the real-world experiments were somewhat lacking (see next section).
- __Manuscript clarity:__ The manuscript is well written and contained sufficient detail.

__Weaknesses:__

- __Real-world experiments baselines (quality):__ The real-world experiments (experiment 6, section 4.5) only compared a base diffusion policy versus the full DynaGuide model. No other baselines were included (quality). This is a relatively weak baseline which does not seem to have any goal conditioning? I have asked for clarification regarding this issue in the questions section.
- __Additional parameter tuning (significance):__ The method modulates between a base diffusion policy and task-based goal arrival (equation 3) weighted by a scaling guidance parameter. Selection of this scaling parameter might require additional tuning to achieve a reasonable policy versus other more direct optimization methods.

__Current Assessment and Suggestions for Improvement:__

I have currently marked this work as borderline accept. I think the submission is of good quality (based on the discussion related to strengths and weaknesses above) however I would of appreciated more baselines comparisons in the real-world experimentation section.

---

> ### Author Rebuttal · Authors · 2025-07-28
>
> Thank you for your detailed and thoughtful review of our paper. We appreciate your recognition that DynaGuide presents a novel approach for steering diffusion policies, and we appreciate your highlighting the performance boosts of DynaGuide in situations with under-represented behaviors and multi-guidance conditions.
>
> Two main points are raised in the weaknesses / questions section, which we will address here, along with other minor points.
>
> # Real-World Experiments: Lack of other Baselines
>
> ## Lack of goal-conditioned baseline
> The primary concern mentioned in the weaknesses section is the **lack of a goal-conditioned baseline in the real robot experiments**. As mentioned in lines 325-326 and lines 71-74, one of the primary purposes of the real robot experiments is to test the hypothesis that DynaGuide can work on *any pretrained policy*. To do this, we use a publicly-accessible off-the-shelf policy [1], which unfortunately does not have a goal-conditioned version. Therefore, in that set of experiments, it was not possible to conduct a fair comparison against a goal-conditioned model, which we would have needed to train ourselves.
>
> On the longer time scale, we can conduct an additional robot experiment set where we train our own policies and allow for goal conditioned comparisons. If we train our own policies with our environment-specific data, we anticipate the results to be stronger across the board.
>
> ## Lack of other baselines (additional experiment)
> The simulation environments provided a stabler and repeatable environment for head-to-head comparisons, which is why we chose the simulation setup to conduct extensive baseline comparisons (exp 1-5). DynaGuide does not make any assumptions that would cause a significant difference between real and simulated experiments. However, we acknowledge the reviewer’s concerns about the need for more baselines in a real-world setup.
>
> We added the DynaGuide-Sampling (GPC) baseline on the real robot. Due to the short rebuttal turnaround time, we were only able to evaluate it on the first real robot experiment (pick red cup and place on black saucer). We got the following results (20 trials per experiment)
>
> | Method              | Red  (steering target) | Grey            |
> |---------------------|------|-----------------|
> | DynaGuide    [from paper]       | 75%  | 25%      |
> | DynaGuide-Sampling [NEW]  | 60%  | 40%             |
> | Base Policy   [from paper]      | 55%  | 45%             |
>
> As noted in the paper, the pretrained base policy is biased towards one side of the environment (i.e., it will typically grab cups on the left side). We shuffle the cups equally for fairness, but to get above ~50%, the method must overcome this bias. Sampling was not effective in removing the bias, while DynaGuide was able to accomplish this.
>
> # Hyperparameter Tuning
> > Selection of this scaling parameter might require additional tuning to achieve a reasonable policy versus other more direct optimization methods.
>
> This is a valid concern, as additional parameters mean additional hyperparameters to tune. However, as seen in Figure 8, the guidance strength parameter (the scaling parameter) is robust to large changes in scale (leftmost plot). DynaGuide is also relatively robust to the sigma parameter (second plot from right) and even the number of guidance conditions (rightmost plot). The stochastic sampling steps (second plot from left) generally increase performance with more steps, so they can be chosen based on resource constraints.
>
> To get the results of DynaGuide shown in the paper, we only needed minimal tuning. Notably, these hyperparameters remain relatively stable across individual tasks within an environment. As an example, here are our optimal hyperparameters for the individual tasks:
>
> | Task         | Scale | Stochastic Sampling | Sigma |
> |--------------|-------|----|-------|
> | switch_on    | 1.5   | 4  | 30    |
> | switch_off   | 1.5   | 4  | 30    |
> | drawer_open  | 1     | 4  | 40    |
> | drawer_close | 1     | 4  | 40    |
> | button_on    | 1     | 4  | 30    |
> | button_off   | 1     | 4  | 30    |
> | door_left    | 1.5   | 4  | 30    |
> | door_right   | 2     | 4  | 15    |
>
> The two most commonly changed parameters are Scale and Sigma. Generally, to deploy DynaGuide, we advise starting with guidance scale 1-2, and increasing if the policy does unwanted behavior. Choose a smaller Sigma if the guidance conditions have more variance and you suspect that only a few are relevant per environment, and vice versa.
>
> # Summary
> Thank you for the comments and feedback. In our response, we detail why our current real-world experiment setup was unable to utilize a goal-conditioned baseline fairly. We conduct an experiment with a GPC baseline on the real robot setup. Finally, we discuss the sensitivity of DynaGuide to its hyperparameters and provide simple rules for tuning them. If any of your concerns were not addressed by this response, we welcome further discussion.
>
> [1] Chi et al. Universal manipulation interface: In-the-wild robot teaching without in-the-wild robots. RSS.

---

> > ### Comment · Reviewer_RYQt · 2025-08-03
> >
> > Thank you for responding to my review. From what I understand, the authors plan to compare against additional real-world baselines in the future ("On the longer time scale, we can conduct an additional robot experiment set where we train our own policies and allow for goal conditioned comparisons") but have added results for the `DynaGuide-Sampling` for the first cup preference real-world experiment. I also understand that "one of the primary purposes of the real robot experiments is to test the hypothesis that DynaGuide can work on any pretrained policy" but this seems a somewhat arbitrary constraint to me. Ideally I would of liked to see the same rigor in experimental baselines applied in simulation carried over to real-world experiments. I sympathize that there may not be enough time to conduct additional testing, however I feel that this is important to the real-world experimental quality of this paper. As such I have left my overall rating unchanged at the moment.
> >
> > Thank you for also providing additional test results over different hyperparameters and illustrating that the optimal hyperparameters found remain relatively constant over different tasks which eases my concerns somewhat.

---

### Note · Authors · 2025-08-12

Thanks to all the reviewers for engaging in very productive discussions that have highlighted DynaGuide’s strengths, suggested additional experiments, and refined its limitations. We provide a brief summary of the discussion:

## Additional Experiments

Multiple reviewers requested additional baselines.

- Reviewer RYQt requested a more extensive set of experiments on the real robot, which we start to provide in our additional experiment with DynaGuide-sampling on the first cup choice task. We show that DynaGuide still outperforms its sampling-based counterpart.
- Reviewer yJvQ & n1oB requested experiments where the dynamics model is trained on the same data split (CALVIN-D) as the base policy. This smaller dataset reduces performance but maintains significance over base policy rates.
- Inspired by Reviewer yJvQ’s comment, we try *adding* data (CALVIN-D -> ABCD) to the goal-conditioned baseline and show that its performance does not increase much, indicating that DynaGuide can use additional data better than a goal-conditioned baseline.
- Reviewer n1oB requested stronger goal-conditioning baselines, which we accomplish by implementing a HER objective. Even with a stronger training protocol, goal conditioning is less successful than DynaGuide on Exp. 3.
- Reviewer n1oB requested classifier-free guidance as a baseline, which we start to provide on the multi-objective setup (Exp 4). We will fully run this baseline on the final publication.


Overall, these additional experiments showed that DynaGuide still maintains performance across simulation and real, and across different data assumptions and baseline training protocols.



## Addressing Questions
We list some major questions below


- Q: Is it hard to find the right parameters? A: DynaGuide is pretty robust to various hyperparameters, especially between tasks in the same environment. (See: response to RYQt)
- Q: Is it slow? A: It is slower than base policy inference, and slightly slower than its sampling counterpart. However, running at 10 Hz control on an RTX3080, it is real-time (See: response to n1oB)
- Q: Is it OK to compare a future state (dynamics output) to a desired end state (guidance condition)? There is extensive discussion in response to n1oB. To briefly summarize: in CALVIN & Square Navigation, we train the dynamics model to predict the end state directly, and on the real robot, we predict future states but use intermediate guidance conditions sampled from within a trajectory.

---

### Decision · Program_Chairs · 2025-09-17

**Decision:**

Accept (poster)

**Comment:**

(a) This paper proposes DynaGuide, which combines action denoising gradients from a pre-trained policy with an inverse dynamics based guidance gradient. This guidance grad increases the probability of meeting guidance goals. The dynamics model is trained to predict future outcomes and compares them to guidance outcomes. The key hypothesis is that this helps the policy in reaching under represented parts of the manifold and also helps when there are multiple objectives to optimize for.

(b) This paper presents a relatively new way of combining classifier style guidance with a learned dynamics model, and moreover it can leverage pre-trained diffusion policies. Multiple simulated tasks and real robot experiments were conducted, with added baselines during rebuttal (HER, classifier-free guidance, dynamics trained only on D split). The paper is also clearly written, includes detailed appendices, ablations, and released code.

(c) Initial submission lacked strong baselines in robot experiments (e.g., goal-conditioned, HER, classifier-free guidance). Some were added in rebuttal, but not fully comprehensively. DynaGuide requires access to goal states (images). In general this may be less feasible than language based specifications.

(d) The paper is technically sound, makes a meaningful/interesting contribution, and provides extensive experiments and rebuttal clarifications. I like that it exploits pre-trained policies and so it has the potential to be generally useful in practice. It is also nice to see this method being applied on real robotics problems. The authors also seemed to have addressed all the major points raised during the rebuttal process.

(e)
- RYQt was concerned about missing real world baselines. Authors added DynaGuide sampling baseline, explained lack of goal-conditioned real robot baseline due to pre-trained policy choice, and discussed robustness of hyper parameters. Reviewer maintained score but acknowledged improvements.

- n1oB raised issues around conceptual novelty, reliance on goal states, missing HER/CFG baselines, and wall-clock overhead. Authors added HER and CFG baselines, clarified guidance assumptions, provided timing comparisons, and discussed applicability to stochastic settings.

- ma6B generally positive, with concerns about reliance on dynamics model, lack of trajectory control, and missing baselines. Authors clarified sensitivity to negative goals, extended discussion on alternative modalities, and added new baselines.